# XTransfer: Modality-Agnostic Few-Shot Model Transfer for Human Sensing at the Edge

## Abstract

Deep learning for human sensing on edge systems presents significant potential for smart applications. However, its training and development are hindered by the limited availability of sensor data and resource constraints of edge systems. While transferring pre-trained models to different sensing applications is promising, existing methods often require extensive sensor data and computational resources, resulting in high costs and poor adaptability in practice. In this paper, we propose `XTransfer`, a first-of-its-kind method enabling modality-agnostic, few-shot model transfer with resource-efficient design. `XTransfer` flexibly uses single or multiple pre-trained models and transfers knowledge across different modalities by (i) *model repairing* that safely mitigates modality shift by adapting pre-trained layers with only few sensor data, and (ii) *layer recombining* that efficiently searches and recombines layers of interest from source models in a layer-wise manner to create compact models. We benchmark various baselines across diverse human sensing datasets spanning different modalities. Comprehensive results demonstrate that `XTransfer` achieves state-of-the-art performance while significantly reducing the costs of sensor data collection, model training, and edge deployment.

## 1 Introduction

Human sensing refers to the process of capturing and interpreting data related to human activities, behaviors, and physiological states using various sensors (Zhang et al., 2023a; Ystgaard et al., 2023). With the proliferation of edge devices equipped with sensors, human sensing plays a vital role in edge systems to understand contexts and enable smart applications, ranging from activity recognition to emotion or vital sign detection (Ahamed & Farid, 2018). Deep learning (DL) offers robust performance in interpreting sensor data, and its deployment on edge devices improves privacy and reduces bandwidth (Zhang et al., 2020; Chen et al., 2020). However, existing DL solutions are often data- and resource-intensive for training and deployment (Dhar et al., 2021), posing significant challenges for data collection and edge deployment in human sensing (Sobin, 2020; Liu et al., 2019).

Unlike other data modalities (*e.g.*, vision or text), collecting human sensing data for training can be costly and even impractical. This is uniquely due to the data that presents brittleness (*i.e.*, noise sensitivity, low SNR), inconsistency (*i.e.*, user variance, scenario changes), and heterogeneity (*i.e.*, differences in hardware configuration) (Lane & Georgiev, 2015; Jeyakumar et al., 2019; Teh et al., 2020). Collecting such data typically requires ethical approvals, as it involves sensitive information (*e.g.*, location, motion patterns) that raises privacy concerns (Zhang et al., 2022). Manual annotation of large volumes of sensor data is labor-intensive and incurs high costs (Song et al., 2023). Importantly, human sensing spans various modalities (*e.g.*, IMU, ultrasound, mmWave radar) and data types (*e.g.*, spectrograms, Doppler profiles, time series), further amplifying the costs of applying DL solutions and naturally leading to cross-modality scenarios (Xiao et al., 2022; Zhang et al., 2023a).

Prior research has explored Few-shot Learning (FSL) (Wang et al., 2024), transfer learning (Dhekane & Ploetz, 2024), and their combination (*i.e.*, cross-domain FSL) (Thukral et al., 2025; Yin et al., 2024) to notably reduce data collection costs by adapting pre-trained models to sensing applications. However, they still rely on large-scale labeled datasets from the same modality to costly train source models from scratch (Song et al., 2023), or face challenges such as *modality shift* and *poor adaptability* when leveraging pre-trained models (Oh et al., 2022; Tao et al., 2022; Guo et al., 2022), particularly in cross-modality settings. Recent advances in multi-modal learning (Yang et al., 2023;

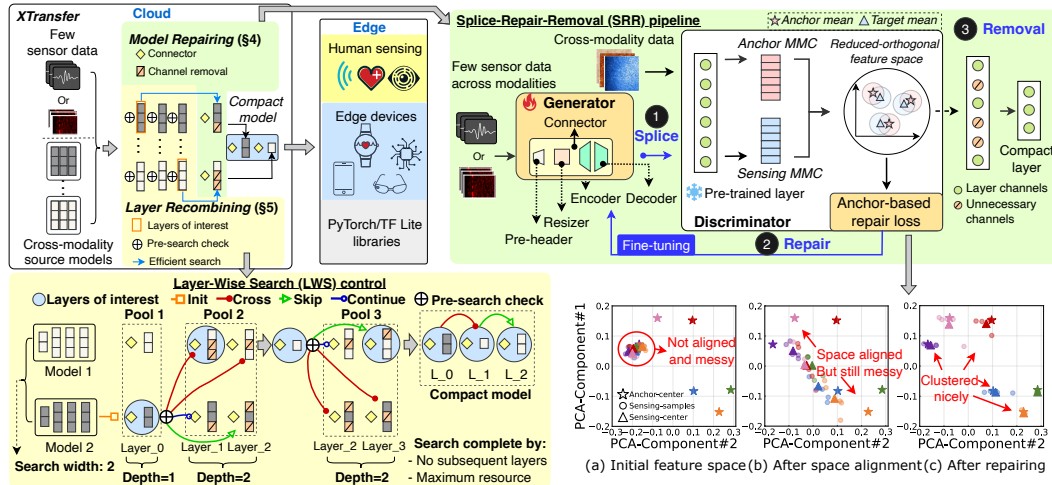

Figure 1: Overview. XTransfer transfers source models across modalities with few sensor data through model repairing (SRR pipeline) and layer recombining (LWS control). LWS control first segments source models into layers and operates layer-wise search across pools. At each pool, the pre-search check decides which layers need repairing, then SRR pipeline repairs them and LWS control selects layers of interest. These layers are incrementally recombined during the search, resulting in a compact model for enabling human sensing at the edge. Subfigures (a)–(c) illustrate the feature space evolution before and after repairing.

Chen & Yang, 2025; mengxi Chen et al., 2024; Madaan et al., 2024; Yang et al., 2024; Baldenweg et al., 2024) (*e.g.*, CLIP-based models (Girdhar et al., 2023; Wu et al., 2023; Moon et al., 2023)) and cross-modal learning (*e.g.*, image-to-sensor distillation (Zhao et al., 2018; Song et al., 2022; Gurbuz et al., 2020; Zhang et al., 2023c) or alignment-based transfer (Kamboj et al., 2025; Chen et al., 2024)), demonstrate the feasibility of knowledge transfer across modalities, but they remain modality-specific (*i.e.*, relying on shared semantic feature spaces), and typically require large-scale paired data with resource-intensive training and deployment, leading to high costs and making them impractical for few-shot adaptation to support human sensing at the edge. These limitations motivate the need for a *scalable* and *adaptable* model transfer that can broadly adapt pre-trained models across modalities (*e.g.*, vision or text) to new sensing modalities (*e.g.*, radar or bio-signals) with few data.

In practice, however, addressing this need is uniquely challenging. Due to substantial differences in data characteristics—such as shape, distribution, and feature representation—between source and target modalities (*i.e.*, *modality shift*), transferred models often suffer from significantly degraded performance (*i.e.*, *negative transfer*) (Guo et al., 2022; Oh et al., 2022). Our experiments reveal that state-of-the-art (SOTA) baselines (Table 1) fail to reach oracle performance (Gong et al., 2019) and suffer from substantial *overfitting* when applied to human sensing tasks with few data, falling notably short of the *goal*, as shown in Figure 2(a). These highlight the underlying challenge of latent feature misalignment across modalities. Also, these methods often neglect *resource constraints*, rendering them impractical for edge deployment, especially in advanced multi-source settings (Zhao et al., 2020; Yue et al., 2021a; Lee et al., 2019). While methods such as neural architecture search (NAS) (Wen et al., 2023; Han et al., 2021), pruning (Frankle & Carbin, 2019; Shen et al., 2022), or quantization (Polino et al., 2018), aim to reduce resource demands, they often fail to maintain performance under modality shift and few data. As shown in Figure 2(c), our experiments using pruning (Shen et al., 2022) demonstrate significant model accuracy loss by 50.5% on average.

Driven by the increasing availability of high-quality pre-trained models from public repositories (*e.g.*, PyTorch Hub (Hub, 2022)), we propose XTransfer, a first-of-its-kind method enabling modality-agnostic, few-shot model transfer. It flexibly leverages public pre-trained models and transfers knowledge across different modalities (*e.g.*, image, text, audio, or sensing) to human sensing tasks using only few sensor data, significantly reducing the costs of large-scale data collection and training from scratch. Its resource-efficient design further optimizes edge deployment, enabling both scalability and adaptability to diverse human sensing applications. XTransfer uniquely repairs, searches and recombines *layers of interest* from pre-trained models under cross-modality FSL settings (see Appendix A), resulting in compact models. To mitigate modality shift and prevent overfitting, it enables *model repairing* using a novel Splice-Repair-Removal (SRR) pipeline (see Section 4). SRR pipeline transforms a few sensor data to broadly adapt pre-trained layers by repairing layer-wise

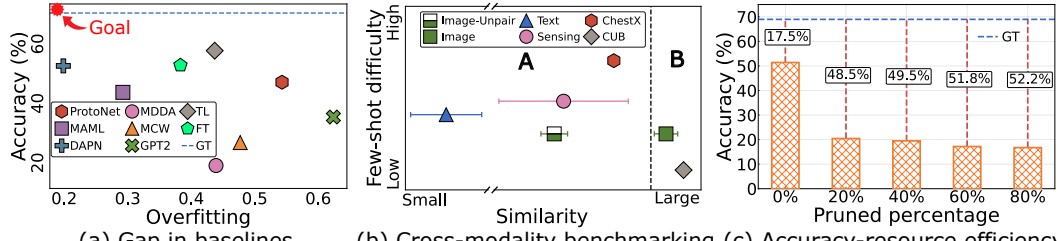

Figure 2: Preliminary study. **(a)** reveal baseline performance gap [1]. **(b)** shows the average similarity and FSL difficulty across all sensing datasets to source modalities (*e.g.*, Image, Text, Sensing) using default reshaping (see Appendix B.1). 2 distinct areas represent similarity levels (A–hard, B–normal). Key findings: **1)** compared to CUB, similarity levels across modalities are notably low, *e.g.*, Text and Sensing fall into Area A, indicating a significant modality shift; **2)** compared to Image-Unpair (*i.e.*, no class pairing) in Area A, Image surprisingly falls into Area B, indicating that pairing classes may enhance cross-modality similarity (motivating our techniques in Section 4.1); **3)** compared to Text and Sensing, Image exhibits more stable standard deviations and lower FSL difficulty, suggesting better potential for broad adaptability. **(c)** shows model pruning bottlenecks.

misalignment and harmonizing latent feature distributions across modalities. `XTransfer` also features *layer recombining* using a novel Layer-Wise Search (LWS) control mechanism (see Section 5) to identify and recombine *layers of interest* by conducting a layer-wise search. Building on the insight that not all repaired layers contribute, LWS control discards such layers and recombines only useful ones to optimize model compactness using principles from NAS (White et al., 2023). To further *accelerate* the search in multi-source settings, a *pre-search check* strategy is designed to efficiently avoid unnecessary repairing.

The main contributions are summarized as: 1) a first-of-its-kind method, `XTransfer`, that enables modality-agnostic, few-shot model transfer with resource-efficient design for human sensing on edge systems, as shown in Figure 1; 2) a novel SRR pipeline that effectively mitigates modality shift and prevents overfitting, harmonizes latent feature distributions across modalities; 3) a novel LWS control mechanism that efficiently searches layers of interest from multiple sources in a layer-wise manner to create compact models; and 4) extensive experimental evaluations (see Section 6) under cross-modality FSL settings, benchmarking SOTA baselines (Table 1) on 8 public datasets with corresponding pre-trained source models across diverse modalities (*e.g.*, image, text, audio, and sensing) and 7 target datasets (Table 2) using 3 commercial edge devices. The results show that `XTransfer` achieves a significantly enhanced accuracy-to-resource (ATR) ratio (see Section 6.1), outperforming single- and multi-source baselines by 1.6 to 29 times and 16.6 to 98 times, respectively, while reducing layer search time by 2.1 to 4 times and substantially lowering the costs of data collection, model training, and edge deployment, demonstrating broad adaptability.

## 2 RELATED WORK

**Learning with few data for human sensing.** Recent works in FSL and cross-domain FSL have proposed *single-* and *multi-source* methods. Single-source methods align latent features between source models and target data via distance-based objectives (Zhao et al., 2021; Yue et al., 2021b), whereas multi-source methods leverage the Wisdom of the Crowd principle through unsupervised source distillation (Zhao et al., 2020; Yue et al., 2021a) or maximal correlation analysis (Lee et al., 2019). Recent efforts have also explored fine-tuning LLMs (*e.g.*, GPT2 (Zhou et al., 2023)) for time-series analysis. However, testing on benchmarks (Liang et al., 2021; Oh et al., 2022), our results reveal that they suffer from severe overfitting under cross-modality FSL settings. LLM-based data augmentation (Leng et al., 2024; 2023) helps reduce data collection costs, but remains modality-specific and orthogonal to our focus. Recent advances introduce foundation models (FM) (Abbaspourazad et al., 2024; Weng et al., 2024) or prompt-based FM adaptation (Li et al., 2025) for human sensing, but these methods assume either a pre-existing target-modality FM or access to large-scale unlabeled or paired data for feature alignment. Existing attempts in human sensing FSL (Gong et al., 2019; Wang et al., 2024; Thukral et al., 2025; Yin et al., 2024) remain constrained by a strong reliance on learning in the same modality, resulting in costly source data collection and

---

[1]We test a pre-trained ResNet18 on miniImageNet as source and HHAR dataset as target under a 5-shot setting (see Appendix A).

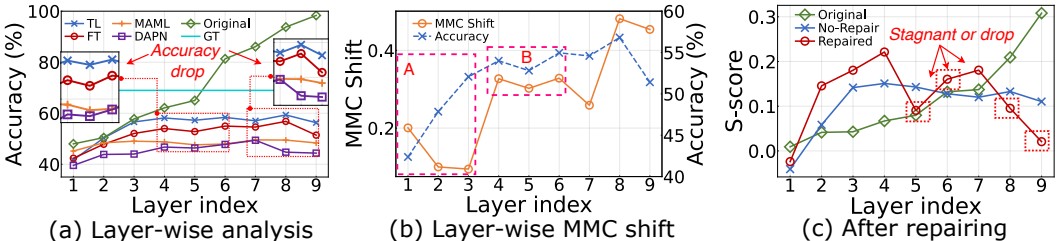

Figure 3: Design insights. **(a)** Layer-wise accuracy convergence using baselines is disrupted due to modality shift. **(b)** In area A, MMC shift stays low and hence accuracy rises, indicating a small latent feature gap. In area B, MMC shift notably increases with layer index, where excessive latent feature deviation begins to reduce accuracy. **(c)** After repairing, layer S-score improves, but stagnation occurs at certain layers.

training. In contrast, `XTransfer` enables layer-wise model repairing to safely mitigate modality shifts using only few sensor data, achieving significant cost reduction with broad adaptability.

**Resource constraints optimization.** Model compression techniques such as model pruning (Frankle & Carbin, 2019; Shen et al., 2022), quantization using low-precision data types (Polino et al., 2018), model merging by fine-tuning shared layers (Padmanabhan et al., 2023), one-size-fits-all model (Cai et al., 2020), and NAS techniques such as model structure optimization (Wen et al., 2023; Han et al., 2021) or search using few data (Eustratiadis et al., 2024; Xu et al., 2022), have shown success in reducing model resource overhead. However, they strongly rely on either sufficient target datasets or minimal modality shifts to maintain model performance. `XTransfer` uniquely applies layer recombining to optimize model structures while strengthening layer-wise dependence, by efficiently searching and recombining layers of interest from pre-trained source models at scale. It not only creates compact models for streamlined edge deployment but also further enhances model repairing.

## 3 PRELIMINARY

**Insights in cross-modality FSL.** We apply a standard cross-domain FSL benchmarking (Oh et al., 2022) to gain insights. It uses Earth Mover's Distance (EMD) to measure similarity between source models (ResNet18 from PyTorch Hub (Hub, 2022)) from 3 modalities, *e.g.*, miniImageNet (Image), Newsgroup (Text), and OPPORTUNITY (Sensing) and targets, including 5 human sensing and 2 well-known image datasets (CUB (Welinder et al., 2010) and ChestX), detailed in Table 2. It also assesses FSL difficulty by fine-tuning source models on targets under 5-shot settings. Since source models involve a large number of classes [2] (*e.g.*, 100 classes in miniImageNet), we propose a class pairing strategy that pairs source and target classes exhibiting similar latent feature distributions across modalities. The key insights are shown in Figure 2(b).

**Layer-wise analysis.** We perform a layer-wise analysis to gain insights, focusing on accuracy changes and latent feature distributions at each layer. Since effective training relies on learning discriminative features (Islam et al., 2021), we benchmark latent feature distribution clustering and use Mean Magnitude of Channels (MMC) (Luo et al., 2022) of each layer as the default metric to quantify layer-wise performance, as it offers a lightweight measure for layer channel importance. We define a single layer or a dependent layer block (*i.e.*, L-Blocks in Table 3)—where layers are not structurally separable—as the default segmentation unit (*i.e.*, *L-unit*).

To set up, we continue to use the ResNet18 (Hub, 2022) on HHAR under 5-shot settings, and segment model into 9 L-Blocks. We also use miniImageNet as source (Original), and examine the single-source baselines and MetaSense (Gong et al., 2019) as oracle, detailed in Table 1. Figure 3(a) shows that negative transfer (Zhang et al., 2023b) disrupts accuracy convergence across layers (*i.e.*, layer-wise dependence), causing layer-wise misalignment and accuracy drops. These misalignments correlate with MMC shifts, where greater shifts correspond to larger accuracy drops shown in Figure 3(b), indicating inconsistent latent feature interpretation during fine-tuning. To track MMC shifts, we essentially employ the Silhouette score (S-score) (Shahapure & Nicholas, 2020), which captures inter- and intra-class distances and reflects the discriminability of layer-wise latent features.

---

[2]The number of classes in the source model should be greater than that in the target.

**Problem statement.** We aim to minimize layer-wise MMC shifts under cross-modality FSL settings, where pre-trained source models across modalities are adapted to different target modalities using only few sensor data. This requires addressing several interconnected challenges that shape our design. **First**, adaptation is hindered by structural and semantic gaps between source and target modalities, where data formats, feature shapes, and class semantics often differ. To address this, we propose the SRR pipeline, which dynamically transforms few sensor data to broadly adapt pre-trained layers through three stages: 1) Splice, which structurally connects the heterogeneous layers; 2) Repair, which aligns layer-wise latent representations across modalities via an anchor-based generative transfer module; 3) Removal, which removes unnecessary layer channels after repair. **Second**, the latent feature space at each layer is highly complex due to the *multidimensional* and *noisy* nature of MMC across source models. This complexity increases with layer depth, as the strong layer-wise dependence formed during source model training becomes increasingly rigid, making it difficult to repair using only few sensor data. To mitigate this, we perform repair in a reduced, orthogonal anchor space, enabling stable layer-wise alignment under few-shot constraints. **Third**, Figure 3(c) shows that S-scores may stagnate or even drop at certain layers (*i.e.*, ineffective layers), impairing overall convergence. It indicates that not all repaired layers contribute positively, motivating our LWS control design to selectively *recombine* layers of interest while discarding others. To ensure efficient and stable search, especially in multi-source settings with an expanded search pool, we design an NAS-inspired search strategy under resource constraints, supported by both pre-search check and dynamic search range mechanism that accelerate and stabilize the search.

## 4  Model Repairing

Enabled by SRR pipeline, this process safely repairs layer-wise misalignment by minimizing MMC shifts, effectively restoring layer performance from source models using few sensing inputs under cross-modality FSL settings. Three stages are designed, as shown in Figure 1. In the Splice stage, default reshaping (see Appendix B.1)—using fixed down/up-sampling—cannot be fine-tuned and may lose important features, especially when layer shapes vary during layer-wise search. We propose a compact *connector* placed between sensing inputs and layers, consisting of a Pre-header (adaptive convolutional layer), Resizer, and one encoder-decoder pair to bridge cross-modality structural gaps. In the Repair stage, each connector is fine-tuned by the *generative transfer module* to generate adaptive layer channels, preserve key sensing features, and minimize MMC shifts. Finally, the Removal stage applies *layer channel removal* (see Appendix B.2) to further enhance repair.

### 4.1  Repairing in complex feature space

Since the original latent features from source models are well-trained for high discriminability, our intuition is to safely fine-tune few sensor data by aligning the misinterpreted latent features with the original features (*i.e.*, anchors) to minimize MMC shifts. We hence propose cross-modality *anchor-based* alignment, which aligns source MMCs as *anchors* (*i.e.*, anchor MMCs) with sensing MMCs. To handle complex feature spaces, our insight is to reduce MMC dimensionality by preserving key layer channels and excluding irrelevant or noisy ones.

**Reduced-orthogonal feature space.** We project anchor MMCs into a lower-dimensional, orthogonal feature space using principal component analysis (PCA) (Valpola, 2015) (*i.e.*, anchor PCA space). Unlike the raw MMC feature space, where layer channel dimensions are often noisy and correlated, PCA enforces orthogonality across components, removing redundancy and suppressing noise. This enables a compact subspace (*i.e.*, defaulting to 2 components) where the projected data captures maximum variance and forms highly clustered distributions. The optimized *component weights* (*i.e.*, projection coefficients) highlight the most important layer channels contributing to layer performance. Given high discriminability of the original layers, the projected anchor MMCs preserve high clustering performance, with class distributions tightly centered (*i.e.*, Anchor-center), as shown in Figure 1(a). Once initialized for each layer, this anchor PCA space is reused to project sensing MMCs.

**Feature space alignment.** After projecting sensing MMCs, we observe that the sample distribution for each class, along with their centroid (*i.e.*, Sensing-center), appears disorganized compared to the well-clustered Anchor-center at each layer, as shown in Figure 1(a). Specifically, the scale and

orientation of Sensing-center are notably different from those of Anchor-center, caused by MMC shifts. To align the scale, we first define a scale function based on the mean shift in inter-class distance $\text{InterD}(\cdot)$, formulated as $S = Mean(InterD(Pro(fs_{ij}))) / Mean(InterD(Pro(ft_{ij})))$, where $Pro(\cdot) = PCA(MMC(\cdot))$ denotes the space project function, $fs$ and $ft$ represent the latent features from both source and sensing modalities, respectively. $i$ denotes the index of pre-trained models, and $j$ denotes the layer index in each model $i$. Next, to align the orientation, we design a rotation alignment function that minimizes the cosine angle (Nguyen & Bai, 2010) by fine-tuning a multi-dimensional rotation matrix $[M_{rot}]$ with dimensions matching the number of PCA components:

$$\arg \min_{[M_{rot}]} Cos(Cen(Pro(fs_{ij})), Cen(Pro(ft_{ij}) * S) \cdot [M_{rot}]) \tag{1}$$

where $Cos(\cdot)$ denotes the cosine similarity function, and $Cen(\cdot)$ calculates the centroid of the projected distribution. Figure 1(b) shows the aligned results.

**Anchor class pairing.** Building on the key finding that class pairing boosts latent feature alignment across modalities (see Section 3), we propose selecting source classes with the highest S-scores as *anchor* classes to pair with sensing classes to facilitate the anchor-based alignment. We minimize pairing shift (*i.e.*, the sum distance across each pair) via a linear sum assignment solved using the standard Hungarian algorithm, outputting the optimal pairing set $pairST$.

### 4.2 REPAIRING LAYER-WISE DEPENDENCE

**Generative transfer module.** To effectively restore layer-wise dependence, we design a *generative transfer module* that enhances the anchor-based alignment through generative learning. It consists of a generator (*i.e.*, connector) and a discriminator—each frozen pre-trained layer paired with its anchor PCA space (Figure 1). Freezing preserves stable anchors and prevents overfitting. For each layer, given the optimal pairing set $pairST$, the generator is fine-tuned using the discriminator's loss to minimize MMC shifts, and uniquely transforms sensor inputs to align with anchors.

**Anchor-based repair loss.** Building on the generative transfer module, we propose an *anchor-based repair loss function* ($loss_{srr}$) to harmonize latent feature distributions across modalities. Given each layer's anchor PCA space, our objective is twofold: 1) *minimize* the distance ($D(\cdot)$) between the centroid of projected anchor MMC distribution and that of the projected sensing MMC distribution for each class pair $(c_s, c_t) \in pairST$ (*i.e.*, positive loss), and 2) *maximize* the distance between sensing MMC samples with different labels ($c_t^-$) (*i.e.*, negative loss). Since the projected anchor MMCs are highly clustered, they can provide the maximum *margin* ($M_{max}$) between anchor clusters. To further improve the negative loss, we update the function by calculating the *anchor-based margin* $margin^c = InterD(Pro(fs_{ij}^c)) - IntraD(Pro(fs_{ij}^c))$, where the $M_{max}$ is selected, and $IntraD(\cdot)$ measures the intra-class distance. Hence, our loss function is formulated as:

$$\begin{aligned}
loss_{srr} =& \frac{1}{N} \sum_{(c_s, c_t) \in pairST}^{N} D(Cen(Pro(fs_{ij}^{c_s})), Cen(Pro(ft_{ij}^{c_t}))) \\
&+ \frac{1}{K} \sum_{(c_t, c_{tk}^-)}^{K} ReLU(M_{max} - D(Pro(ft_{ij}^{c_t}), Pro(ft_{ij}^{c_{tk}^-})))
\end{aligned} \tag{2}$$

where $K$ denotes the number of negative samples, and $N$ denotes the number of classes. The $ReLU(\cdot)$ is used to trigger the negative loss in the condition where the distance between the samples is less than $M_{max}$. Figure 1(c) shows the effectiveness of model repairing.

## 5 LAYER RECOMBINING

Working in conjunction with SRR pipeline for repairing each layer, XTransfer leverages *layer recombining* enabled by LWS control to: 1) perform the pre-search check that accelerates the search process, and 2) effectively select and recombine layers of interest to form a compact model.

### 5.1 SEARCH CONTROL DESIGN

Following NAS principles (White et al., 2023), poor design choices (*e.g.*, search space, actions, strategy) can disrupt the balance between model performance and resource efficiency. As layers of

interest grow, resource overhead can increase exponentially, making dynamic search control essential under on-device resource constraints. To effectively manage, we define the search space as all layers from source models, structured into *search pools* containing layer candidates (*i.e.*, repaired layers), each with width $I$ (number of models) and depth $J$ (number of L-units), forming a window of size $I \times J$. Search actions (*init*, *continue*, *skip*, *cross*) are defined to guide layer recombination within each designated search pool. *Init* selects the starting layer with a default fixed depth of 1. *Continue* and *skip* guide intra-model layer recombination, while *cross* enables inter-model layer recombination.

**Search strategy under resource constraints.** To effectively control search actions within each search pool, we propose a layer value function based on the S-score $Score(\cdot)$ to evaluate each repaired layer performance and the convergence between adjacent layers after each SRR process $SRR(\cdot)$. It is formulated as $V_{ij} = Score(SRR(Pro(ft_{ij}), MT_{ij}, L_{ij}), Y_t))$, where $MT_{ij}$ denotes the connector and $L_{ij}$ represents each repaired layer. Under resource constraints, only the highest-valued repaired layer in each pool is selected. LWS control then checks whether its S-score exceeds that of the previous layer to maintain layer-wise dependence. If successful, the layer $L_{ij}$ is identified as the layer of interest and recombined, and the next pool window begins at layer index $j + 1$. Otherwise, the current pool is discarded, and the search window shifts forward. Considering the constraints, we design a resource-constrained function that works in conjunction with $V_{ij}(\cdot)$, which is formulated as $RC(n) = exp(n/(L-2) - 2) + 1, n \in [1, L]$, where $n$ denotes the current recombined layer index, and $L$ denotes the maximum number of L-units. Each layer's adjusted overhead, $R(n)_{ij}$, is computed as $R(n)_{ij} = (Res(L_{ij}) * RC(n))/Max(Res(P))$, where $Res(\cdot)$ calculates actual cost (*e.g.*, FLOPs), and $Max(Res(P))$ is the peak cost in current pool $P$. The resource-constrained layer value function becomes $VR_{ij} = V_{ij} / R(n)_{ij}$. The objective is defined as $\arg\max_{L_{ij} \in P} VR_{ij}, \ s.t. \sum_0^n Res(L_{ij}) \leq R_{Max}$, where $R_{Max}$ is device-specific resource budget.

## 5.2 EFFICIENT SEARCH

**Pre-search check.** LWS control is designed to handle large search spaces from deep or multi-source models. While fine-tuning the connectors in SRR pipeline is efficient, the large search space can considerably slow down LWS control. We hence propose the *pre-search check* strategy to avoid unnecessary repair steps *before* applying SRR. We track S-score progression across three states—after anchor pairing (Anchor) (*i.e.*, repairing objective), before SRR (Before), and after SRR (After) (*i.e.*, repaired). SRR pipeline consistently improves S-scores from Before to After, approaching the Anchor level, indicating a strong correlation. Based on this, our intuition is to estimate repaired S-score (*i.e.*, repair rate) of each layer based on Before and Anchor states to decide whether to bypass unnecessary repairs. Experiments show the actual repair rate grows exponentially with the recombined layer index (see Appendix B.4). We hence design a *repair rate growth model* formulated as $rate_n^{est} = exp(a * n) + b, n \in [1, L]$, where $a$ and $b$ are optimized via non-linear regression to minimize MSE to fit $rate_n^{est}$, with the objective function defined as $\arg\min_{a,b} \frac{1}{n'} \sum_{n=0}^{n'} (rate_n^{est} - rate_n)^2$, where $n'$ is the number of recombined layers. The estimation begins after *init*. Once a new layer is selected, LWS control updates the estimation.

**Dynamic search range mechanism.** Based on the estimated repair rate, LWS control can initially use a small, fixed search range (*i.e.*, top-1 layer with the highest estimated rate) to efficiently filter out most layer candidates *before* initiating SRR processing. However, because the updates to $rate_n^{est}$ are limited, the estimation may be inaccurate, potentially affecting selection decisions (*e.g.*, missing the optimal layer candidate). To optimize the search stability, we develop a dynamic search range mechanism that adjusts the search range to balance accuracy and efficiency. This mechanism is formulated as $range_n = (1 + \hat{u}^{\pm} rate_n^{est}) * range_{n-1}, \ range_n \in [0.2, 1]$, where the range starts at 0.5 and adjusts based on the previous range. Here, $\hat{u}$ denotes a signed unit vector. If $rate_{n-1}^{est}$ is less than $rate_{n-1}$, it indicates that the estimation might be inaccurate, and $\hat{u}$ will be positive to increase the search range, allowing more layers to be repaired to stabilize accuracy. Otherwise, the range will be reduced to improve search efficiency by bypassing unnecessary repairs.

## 6 EXPERIMENT

### 6.1 EXPERIMENT SETUP

Table 1: State-of-the-art (SOTA) baselines.

| Baselines | Group | Description |
|---|---|---|
| Transfer Learning (TL) (Jiang et al., 2022) | Single-source | A standard transfer learning method that reuses a source model by **freezing all layers**. |
| Fine-tuning (FT) (Oh et al., 2022) | Single-source | A conventional fine-tuning method that **updates all layers** of a pre-trained model using available target data. |
| ProtoNet (Chen et al., 2019) | Single-source | A distance-based FSL method that constructs **class prototypes** by averaging support set and classifies query set in the feature space. |
| MAML (Chen et al., 2019) | Single-source | A gradient-based meta-learning method that learns a source model with **shared initialization parameters** from scratch. |
| DAPN (Zhao et al., 2021) | Single-source | A cross-domain FSL method that uses **adversarial ProtoNet** with a domain discriminator to reduce feature distribution shift. |
| SemiCMT (Chen et al., 2024) | Single-source | A cross-modal learning method that applies **contrastive self-supervised learning** during model adaptation. |
| GPT2 (Zhou et al., 2023) | Single-(LLM) | A LLM-based method that enables **in-context learning** after fine-tuning on few target data. |
| MDDA (Zhao et al., 2020) | Multi-source | A multi-domain FSL method that uses **adversarial discriminative adaptation** and **multi-source knowledge distillation**. |
| MCW (Lee et al., 2019) | Multi-source | A multi-domain FSL method that uses **maximal correlation analysis** to fuse multiple source domains for few-shot model adaptation. |
| MetaSense (Gong et al., 2019) | Oracle-(GT) | An advanced FSL method based on MAML that uses **extensive target data** to train from scratch, serving as an oracle upper bound. |

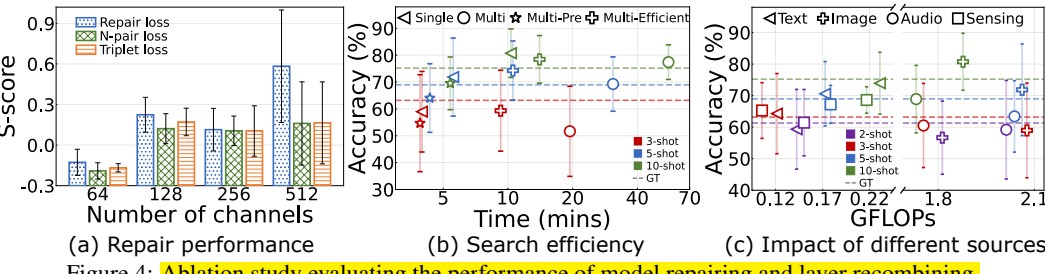

Figure 4: Ablation study evaluating the performance of model repairing and layer recombining.

(a) Repair performance  (b) Search efficiency  (c) Impact of different sources

We benchmark XTransfer against SOTA baselines, grouped into single- and multi-source methods (Table 1). Evaluation is conducted on 8 widely-used public datasets under cross-modality FSL settings (Table 2), using their corresponding public pre-trained models as sources, from diverse modalities (*e.g.*, image, text, audio, and sensing) with various backbones such as ResNet18 from PyTorch Hub (Hub, 2022) (Table 3). To evalu-

| Dataset | Modality | Subject | Class | Train Size | Type | Shape | Privacy |
|---|---|---|---|---|---|---|---|
| **Source** | | | | | | | |
| miniImageNet (Vinyals et al., 2016) | Image | - | 100 | 50k | - | 3x84x84 | Public |
| miniDomainNet (Zhou et al., 2021) | Image | - | 126 | 137.5k | - | 3x224x224 | Public |
| Office-31 (Saenko et al., 2010) | Image | - | 31 | 3.3k | - | 3x224x224 | Public |
| Office-Home (Venkateswara et al., 2017) | Image | - | 65 | 12.7k | - | 3x224x224 | Public |
| Caltech-101 (Fei-Fei et al., 2004) | Image | - | 101 | 7.3k | - | 3x224x224 | Public |
| Newsgroup (Thulasidasan et al., 2021) | Text | - | 20 | 15k | - | 100x1x1000 | Public |
| VoxCeleb (Nagrani et al., 2017) | Audio | - | 100 | 14.9k | Time-S | 1x512x300 | Public |
| OPPORTUNITY (Roggen et al., 2010) | IMU | 12 | 11 | 3.7k | Time-S | 77x1x100 | Public |
| **Target** | | | | | | | |
| HHAR (Stisen et al., 2015) | IMU | 9 | 6 | - | Time-S | 6x1x256 | Public |
| WESAD (Schmidt et al., 2018) | ECG | 15 | 3 | - | Time-S | 10x1x256 | Public |
| Gesture | IMU/PPG | 10 | 8 | - | Time-S | 7x1x200 | Private |
| Writing | Ultrasound | 10 | 10 | - | Time-S | 1x1x256 | Private |
| Emotion | mmWave | 10 | 7 | - | Spectrum | 3x224x224 | Private |
| BP | mmWave | 10 | - | - | Doppler | 1x1x499 | Private |
| ChestX (Wang et al., 2017) | Image | - | 5 | - | - | 1x1024x1024 | Public |

Table 2: Source and target datasets specifications.

ate XTransfer, we develop 4 real-world applications and build testbeds on 3 commercial edge devices (*e.g.*, smartwatch, smartphone, Raspberry Pi, see Appendix Table 7 for details).

Based on the testbeds, we collect 4 human sensing datasets (marked as Private in Table 2) as targets, from 40 participants in real-world settings, with ethics approval from the Human Research Ethics Committee of our institute. Additionally, we include 2 public sensing datasets and 1 image dataset as targets for standard benchmarking. We apply standard Leave-One-Out

| Backbone | Layers | L-Blocks | GFLOPs | Params | Kernel | Shape |
|---|---|---|---|---|---|---|
| ResNet18 (Chen et al., 2019) | 18 | 9 | 3.67 | 11.18M | 2D | 3x224x224 |
| ResNet18-1D (Chen et al., 2019) | 18 | 9 | 0.39 | 3.9M | 1D | 100x1x1000 |
| Multi-Conv4 (Zhao et al., 2020) | 4x5 | 4 | 0.3 | 1.15M | 2D | 3x32x32 |
| Multi-ResNet10 (Zhao et al., 2020) | 10x5 | 5 | 0.7 | 24.5M | 2D | 3x84x84 |
| Multi-ResNet18 (Zhao et al., 2020) | 18x5 | 9 | 18.35 | 55.9M | 2D | 3x224x224 |
| Conv4 (Chen et al., 2019) | 4 | 4 | 0.02 | 0.07 | 1D | 1x1x500 |
| Conv5 (Gong et al., 2019) | 5 | 5 | 0.19 | 30.42M | 1D | By Input |
| GPT2 (Zhou et al., 2023) | 4x6 | 6 | 0.41 | 82.9M | 1D | By Input |

Table 3: DL model backbone specifications.

Cross-Validation (LOOCV) (Gong et al., 2019) for evaluating human sensing datasets, and adopt accuracy-to-resource (ATR) ratio, defined as $ATR = Accuracy \, / \, (\alpha Norm(FLOPs)) + (1 - \alpha)Norm(Params))$ for evaluating resource-accuracy efficiency, where $\alpha$ is set to 0.5 by default (see Appendix A for more details on our experiment setup).

## 6.2 XTRANSFER PERFORMANCE

**Repair performance.** We first evaluate how our Repair loss reduces MMC shift to enhance the S-score of each layer with different MMC dimensionalities in the SRR pipeline. We use HHAR and the standard pre-trained ResNet18 on miniImageNet in a 5-shot setting. The layer channel removal is disabled in this experiment. We test with 4 different levels of MMC dimensionality (*i.e.*, number of layer channels) ranging from 64 to 512. We employ N-Pair loss (Zhang et al., 2023c) and Triplet loss (Weller et al., 2022) in FSL sensing tasks as baselines. We run 15 rounds for each setting by default throughout the evaluation. Figure 4(a) presents that our Repair loss consistently achieves the highest

S-score of 0.2 on average, outperforming both N-Pair loss (0.05) and Triplet loss (0.07), across all levels of MMC dimensionality. It indicates that Repair loss effectively enhances layer performance.

**Efficient search performance.** We now evaluate LWS control performance regarding model accuracy and layer search efficiency. We employ HHAR as target, ResNet18 on miniImageNet and Multi-ResNet18 on 5 image datasets as sources shown in Table 2, with a default *search depth* of 3 in 3-, 5- and 10-shot settings. We also utilize MetaSense in Table 1 as the oracle baseline. We compare different configurations: ResNet18 (Single) and Multi-ResNet18 (Multi) without efficient search enabled, Multi-ResNet18 (Multi-Pre) with only pre-search check using top-1 layer selection enabled, and Multi-ResNet18 (Multi-Efficient) with both pre-search check and dynamic search range enabled. As shown in Figure 4(b), Multi-Pre achieves a lowest search time on average across all n-shot settings, but it leads to degraded accuracy by 3.36% and 7.94% on average, respectively, compared to Multi and Multi-Efficient. It also fails to reach oracle-level accuracy and presents the highest standard deviation on average. The results indicate that relying solely on the pre-search check introduces search stability issues due to inaccurate or drifted repair value estimation. In contrast, Multi-Efficient provides a strong balance between accuracy and efficiency. It reduces search time by 2.1 to 4 times compared to Multi in 5- and 10-shot settings, while outperforming the oracle baseline. While Multi-Efficient takes a longer search time than both Single and Multi-Pre, it successfully achieves superior accuracy and search stability, highlighting the benefits of using multiple source models and dynamic search range mechanism. Notably, even the search space is expanded 5 times more compared to Single, Multi-Efficient requires only 2.1 times more search time, indicating strong scalability enabled by the efficient search. The results also reveal that both Multi-Efficient and Multi achieve lower standard deviation on average compared to Single, suggesting that using multiple source models offers commendable search stability. In short, the results prove that the proposed efficient search significantly accelerates the layer search process while improving accuracy and stability.

**Impact of different sources.** Since the selection of pre-trained source models plays a crucial role in `XTransfer`, we evaluate how different source models from various modalities affect the accuracy and resource overhead of the output models. We use HHAR, 2 pre-trained ResNet18s on both miniImageNet (Image) and VoxCeleb (Audio), and 2 pre-trained ResNet18-1Ds on both Newsgroup (Text) and OPPORTUNITY (Sensing), as shown in Table 2. We keep the other experimental setups unchanged and include 2-shot settings. Figure 4(c) shows that the output models using Image source achieve the highest accuracy of 70.5% on average in 3- to 10-shot settings, compared to that of 69.6%, 64.3%, and 67% achieved by using Text, Audio, Sensing sources, respectively. This suggests that the latent features learned from Image, trained on larger datasets with broader class semantics, provide higher discriminability than those learned from other modalities. Notably, Image achieves lower average accuracy in 2- and 3-shot settings, suggesting that extremely low target data may not fully represent the test data distribution, leading to less stable alignment when using Image source. In contrast, both Text and Sensing result in higher average accuracy and reach the oracle, indicating that performance varies with source model quality and semantic relevance to the target. These results also show that `XTransfer` can flexibly reuse diverse source models across modalities to improve ultra low-shot performance. Moreover, the output models using Text or Sensing sources require only 0.16G FLOPs on average, as 1D kernels require much less computation than 2D kernels. Since `XTransfer` prioritizes accuracy, we select Image source models by default. More results see Appendix C.

### 6.3 COMPREHENSIVE COMPARISON

**Cross-modality FSL performance.** We evaluate the overall performance of `XTransfer` regarding both accuracy and ATR across 5 sensing datasets and 1 image dataset as targets (Table 2). We compare `XTransfer` against a range of SOTA baselines shown in Table 1. We also mark our methods as Our-Single and Our-Multi, using pre-trained ResNet18 on miniImageNet and Multi-ResNet18 with 5 corresponding image datasets as sources. In addition, we evaluate the fine-tuning performance of GPT2-small for comparison and align the parameters with the work (Zhou et al., 2023). All methods on target datasets are evaluated under the same cross-modality FSL settings (see Appendix A.1), where only few target data are paired with the sources. MetaSense (Gong et al., 2019) serves as the oracle, trained on Conv5 with sufficient target sensor data (Table 3). For consistency, we also train the

| Method | HHAR (%) | | | WESAD (%) | | | Gesture (%) | | | Writing (%) | | | Emotion (%) | | | ChestX (%) | | |
|---|---|---|---|---|---|---|---|---|---|---|---|---|---|---|---|---|---|---|
| | 3-shot | 5-shot | 10-shot | 3-shot | 5-shot | 10-shot | 3-shot | 5-shot | 10-shot | 3-shot | 5-shot | 10-shot | 3-shot | 5-shot | 10-shot | 3-shot | 5-shot | 10-shot |
| MetaSense (Gong et al., 2019) | 63.2 | 69.0 | 75.2 | 55.1 | 64.0 | 68.3 | 66.3 | 73.4 | 79.4 | 72.1 | 83.3 | 89.6 | 55.5 | 56.3 | 65.5 | 24.8 | 28.1 | 31.7 |
| ProtoNet (Chen et al., 2019) | 50.6 | 45.7 | 49.8 | 43.9 | 49.3 | 52.0 | **50.8** | 55.8 | 59.2 | 69.5 | 78.7 | 73.4 | **51.3** | 49.2 | 54.8 | 21.7 | 23.4 | 24.5 |
| DAPN (Zhao et al., 2021) | 48.7 | 51.2 | 54.7 | 56.9 | 61.2 | 63.1 | 45.8 | 49.4 | 50.5 | **75.1** | 78.6 | 79.6 | 48.0 | 50.0 | **58.7** | 24.0 | 24.4 | 25.5 |
| MAML (Chen et al., 2019) | 36.6 | 42.3 | 42.4 | 39.6 | 57.9 | 65.6 | 37.2 | 41.3 | 39.4 | 31.6 | 39.7 | 35.2 | 22.3 | 26.9 | 31.1 | 22.5 | 20.4 | 21.9 |
| SemiCMT (Chen et al., 2024) | 33.3 | 38.9 | 48.9 | 37.3 | 40.0 | 44.4 | 23.5 | 34.4 | 42.5 | 27.3 | 38.6 | 48.0 | 24.0 | 33.6 | 46.7 | 25.3 | 25.6 | 28.0 |
| GPT2 (Zhou et al., 2023) | 41.0 | 34.0 | 45.0 | 48.0 | 35.0 | 36.0 | **49.0** | 54.0 | 63.0 | 52.0 | 71.0 | 75.0 | - | - | - | - | - | - |
| Our-Single | **59.0** | **71.8** | **80.7** | **77.9** | **78.4** | **81.8** | 45.2 | **69.6** | **77.8** | 72.8 | **87.0** | **91.3** | **43.6** | **55.6** | **61.4** | **27.7** | **28.6** | **30.4** |
| MDDA (Zhao et al., 2020) | 13.2 | 17.7 | 18.8 | 40.0 | 39.3 | 41.9 | 11.2 | 13.9 | 15.2 | 12.5 | 10.9 | 9.0 | 14.4 | 15.3 | 16.0 | 20.6 | 19.6 | 19.9 |
| MCW (Lee et al., 2019) | 27.6 | 25.4 | 26.9 | 30.7 | 32.7 | 32.4 | 22.3 | 23.4 | 25.6 | 32.3 | 31.6 | 30.2 | 25.3 | 27.2 | 26.5 | 22.4 | 21.5 | 22.8 |
| Our-Multi | **59.3** | **74.3** | **78.4** | **76.4** | **77.8** | **78.7** | 48.7 | **73.1** | **78.6** | 74.1 | **86.1** | **90.4** | 42.3 | **55.1** | 58.6 | **29.0** | **30.0** | 30.2 |

Table 4: Comparison in model accuracy. Top-2 results are highlighted in **bold**.

| Method | HHAR | | | WESAD | | | Gesture | | | Writing | | | Emotion | | | ChestX | | |
|---|---|---|---|---|---|---|---|---|---|---|---|---|---|---|---|---|---|---|
| | 3-shot | 5-shot | 10-shot | 3-shot | 5-shot | 10-shot | 3-shot | 5-shot | 10-shot | 3-shot | 5-shot | 10-shot | 3-shot | 5-shot | 10-shot | 3-shot | 5-shot | 10-shot |
| ProtoNet (Chen et al., 2019) | 0.51 | 0.46 | 0.50 | 0.44 | 0.49 | 0.52 | 0.51 | 0.56 | 0.59 | 0.69 | 0.79 | 0.73 | 0.51 | 0.49 | 0.55 | 0.22 | 0.23 | 0.25 |
| DAPN (Zhao et al., 2021) | 0.49 | 0.51 | 0.55 | 0.57 | 0.61 | 0.63 | 0.46 | 0.49 | 0.51 | 0.75 | 0.79 | 0.80 | 0.48 | 0.50 | 0.59 | 0.24 | 0.24 | 0.26 |
| MAML (Chen et al., 2019) | 0.37 | 0.42 | 0.42 | 0.40 | 0.58 | 0.66 | 0.37 | 0.41 | 0.39 | 0.32 | 0.40 | 0.35 | 0.22 | 0.27 | 0.31 | 0.22 | 0.20 | 0.22 |
| SemiCMT (Chen et al., 2024) | 0.33 | 0.39 | 0.49 | 0.37 | 0.40 | 0.44 | 0.24 | 0.34 | 0.43 | 0.27 | 0.39 | 0.48 | 0.24 | 0.34 | 0.47 | 0.25 | 0.26 | 0.28 |
| GPT2 (Zhou et al., 2023) | 0.11 | 0.09 | 0.12 | 0.13 | 0.09 | 0.09 | 0.13 | 0.14 | 0.17 | 0.14 | 0.19 | 0.20 | - | - | - | - | - | - |
| Our-Single | **1.42** | **1.57** | **2.09** | **2.76** | **2.61** | **2.51** | **1.55** | **1.90** | **1.76** | **2.11** | **1.88** | **1.84** | **0.82** | **1.07** | **1.26** | **0.83** | **1.04** | **1.24** |
| MDDA (Zhao et al., 2020) | 0.03 | 0.04 | 0.04 | 0.08 | 0.08 | 0.08 | 0.02 | 0.03 | 0.03 | 0.03 | 0.02 | 0.02 | 0.03 | 0.03 | 0.03 | 0.04 | 0.04 | 0.04 |
| MCW (Lee et al., 2019) | 0.06 | 0.05 | 0.05 | 0.06 | 0.07 | 0.06 | 0.04 | 0.05 | 0.05 | 0.06 | 0.06 | 0.06 | 0.05 | 0.05 | 0.05 | 0.04 | 0.04 | 0.05 |
| Our-Multi | **1.71** | **1.47** | **1.64** | **3.00** | **2.70** | **2.75** | **1.20** | **1.25** | **1.63** | **1.93** | **1.96** | **1.53** | **0.96** | **1.23** | **1.73** | **0.68** | **1.24** | **0.83** |

Table 5: Comparison in ATR ratio. Top-2 results are highlighted in **bold**

| Method | HHAR (%) | | | WESAD (%) | | | Gesture (%) | | | Writing (%) | | | Emotion (%) | | | ChestX (%) | | |
|---|---|---|---|---|---|---|---|---|---|---|---|---|---|---|---|---|---|---|
| | 3-shot | 5-shot | 10-shot | 3-shot | 5-shot | 10-shot | 3-shot | 5-shot | 10-shot | 3-shot | 5-shot | 10-shot | 3-shot | 5-shot | 10-shot | 3-shot | 5-shot | 10-shot |
| DAPN (Zhao et al., 2021) | 48.85 | 49.19 | 51.79 | 44.62 | 49.49 | 48.81 | 42.73 | 48.71 | 50.59 | 43.88 | 56.63 | 57.51 | 44.67 | 59.73 | 63.12 | 46.66 | 47.32 | 48.48 |
| MAML (Chen et al., 2019) | 59.59 | 79.32 | 89.72 | 52.58 | 70.27 | 77.52 | 53.11 | 66.75 | 73.21 | 72.56 | 85.47 | 92.22 | 63.39 | 72.78 | 75.01 | 44.52 | 55.46 | 61.36 |
| SemiCMT (Chen et al., 2024) | 55.60 | 73.43 | 78.91 | 38.83 | 50.33 | 67.89 | 57.02 | 72.16 | 98.04 | 75.14 | 79.12 | 81.79 | 60.30 | 63.43 | 68.60 | 41.54 | 50.05 | 63.43 |
| Our-Single | **18.27** | **18.90** | **19.48** | **15.95** | **16.46** | **16.98** | **20.67** | **21.71** | **22.51** | **19.92** | **20.67** | **21.08** | **18.24** | **19.20** | **19.96** | **16.97** | **17.58** | **18.80** |
| Our-Multi | **18.31** | **20.77** | **22.06** | **18.00** | **19.42** | **23.10** | **20.49** | **22.49** | **24.49** | **18.68** | **19.25** | **21.69** | **16.45** | **18.18** | **19.85** | **15.33** | **17.69** | **19.62** |

Table 6: Comparison in GPU utilization of each method during training. Top-2 results are highlighted in **bold**

oracle using ResNet18 on the Emotion dataset to match 2D kernels. As shown in Table 4 and 5, both our methods outperform the baselines regarding model accuracy and ATR, and successfully reach or surpass the oracle in model accuracy across all target datasets in 5- and 10-shot settings. Specifically, both Our-Single and Our-Multi achieve a significantly higher ATR on average, surpassing the existing single- and multi-source baselines by 1.6 to 29 times and 16.6 to 98 times, respectively. Although our method achieves slightly lower accuracy than the oracle on HHAR, Gesture, and Emotion in 3-shot settings, XTransfer remains flexible to enable further gains by reusing additional source models with higher quality or closer semantic relevance, as suggested by our analysis of source model impact. In summary, XTransfer achieves SOTA accuracy while substantially reducing data and resource costs, providing a scalable and adaptable method for human sensing at the edge.

**Training resource usage.** We also examine the training resource usage of XTransfer by measuring GPU utilization (Util) in the cloud. We focus on the above baselines that require model re-training or adaptation for transfer, *e.g.*, MAML (Chen et al., 2019), DAPN (Zhao et al., 2021), and SemiCMT (Chen et al., 2024)), excluding methods that rely solely on fine-tuning or distance-based techniques. The experimental setup remains the same as above. As shown in Table 6, both our methods achieve the lowest GPU Util across all n-shot settings compared to baselines. Notably, both Our-Single and Our-Multi achieve a substantial reduction in GPU Util by 2.6 to 3.6 times and 2.5 to 3.5 times on average, respectively, compared to baselines. This is due to our layer-wise method, which progressively repairs and recombines only a subset of layers rather than training the entire model at once. It also highlights the effectiveness of our compact connector design. Despite leveraging multiple source models, Our-Multi shows no significant increase (*e.g.*, only 3.67%) in training resource usage, further highlighting its efficiency and scalability. Hence, XTransfer significantly reduces cloud resource usage, enabling scalable model transfer.

## 7 CONCLUSION

This paper presents XTransfer, a novel method that enables modality-agnostic few-shot model transfer with resource-efficient design for human sensing applications. XTransfer harnesses the power of pre-trained models as free sources and requires only few sensor data to create compact models. It significantly reduces the costs associated with sensor data collection, model training, and edge deployment, offering a scalable and adaptable method for advancing human sensing.

**Ethics Statement**  We have reviewed the ICLR Code of Ethics and ensured that this research complies with all relevant rules. Our experiments involve human subjects for sensing data collection, and we obtained approval from the Human Research Ethics Committee of our institution (Reference No: 520221141241381), in accordance with institutional and national requirements. This information is reported in the paper (see Section 6.1).

The paper includes the instructions provided to participants, relevant screenshots of the experimental setup, and information on compensation (if applicable). All participants were volunteers who provided written informed consent before participating. They were given clear instructions regarding the study, its purpose, and any potential risks. Identified risks (*e.g.*, privacy concerns) were minimal and were disclosed to participants through an informed consent process.

The collected data were anonymized at the time of recording, and no personally identifiable information was stored. Data are securely stored and will only be shared under controlled conditions to protect participant privacy.

**Reproducibility Statement**  We have taken extensive measures to ensure the reproducibility of our work. The core method is described clearly in the main paper, with additional technical details provided in Appendix B. Data preprocessing procedures, dataset statistics, hyperparameter and optimizer settings, and complementary information for reproducibility are provided in Appendix A.

We also report the specifications of our computational environment in Appendix A, including hardware setups (CPUs, GPUs, and RAM), memory consumption, and software dependencies. These details are provided to support transparency and reproducibility across different computational platforms.

In addition, we have provided supplementary material to further support the reproducibility. Upon acceptance, we will release our source code in a public GitHub repository, accompanied by detailed instructions to guide researchers in reproducing the experimental results. The repository will include scripts for training and testing, configuration files for the experiments, and documentation of dataset usage. For our proprietary datasets involving human subjects, release will be subject to ethical approval, while all public datasets used in this work have been properly cited and linked.

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

# APPENDIX

The appendix is organized as follows:

- Section A illustrates more details on our experiment setup, including implementation details, developed real-world sensing applications, experimental setups, dataset statistics, evaluation metrics, and computation resources we used for our experiments.
- Section B introduces additional technique details of different components in `XTransfer`.
- Section C gives more experimental results about ablation study and benchmarking.
- Section D describes the usage of LLMs.

## A EXPERIMENT SETUP DETAILS

### A.1 SETTINGS FOR LEARNING WITH FEW DATA

A typical FSL method consists of two stages–meta-training and meta-testing (Chen et al., 2019). In meta-training stage, FSL requires a large-scale source dataset from the same modality (*i.e.*, typically specific to the application) to train source models from scratch. The source dataset is divided into a support set and a query set [3]. Data for each set is loaded according to the *n-way* (*i.e.*, number of classes) and *n-shot* (*i.e.*, number of samples per class) format (Song et al., 2023). The meta-testing stage follows the same structure but uses unseen target data. Specifically, the target support set is used to *adapt* the source models, which are then evaluated on the target query set. Leave-One-Out Cross-Validation (LOOCV) is widely used as the *de facto* standard for evaluating datasets involving multiple users in human sensing (Gong et al., 2019). LOOCV works by leaving one user out as the target for meta-testing while using the remaining users to train the source models during meta-training in each round. The MetaSense (Gong et al., 2019) is used as an oracle baseline under FSL settings, where sufficient target sensing datasets are provided during meta-training stage. Unlike FSL, cross-domain FSL focuses primarily on meta-testing and transfers pre-trained source models across domains (*i.e.*, different users), while remaining within the same modality (Oh et al., 2022). In this work, the cross-modality FSL settings are defined similarly to cross-domain FSL, but extend it to transferring pre-trained source models across different modalities, using only few labeled target data and without assuming any additional unlabeled target data.

### A.2 SETUPS IN PRELIMINARY STUDY

As shown in Figure 2(a), we test a standard *overfitting* metric, defined as $\frac{|Accuracy_{train} - Accuracy_{test}|}{Accuracy_{train}}$ (López et al., 2022). In Figure 2(b), we test 3 source models (using a pre-trained ResNet18 from PyTorch Hub (Hub, 2022)) from 3 modalities, *e.g.*, miniImageNet (Image), Newsgroup (Text), and OPPORTUNITY (Sensing), shown in Table 2. We also use 5 sensing datasets (HHAR, WESAD, Writing, Emotion, and Gesture) as targets, and employ 2 well-known image datasets (CUB and ChestX) as reference targets to the Image, detailed in Table 2. We use a default 5-shot cross-modality FSL setting with LOOCV enabled.

### A.3 SETUPS IN LAYER-WISE ANALYSIS

In Section 3, we conduct a layer-wise analysis to gain insights into cross-modality model transfer with few sensor data. To set up, we continue to use the standard ResNet18 (Hub, 2022) and segment it as 9 L-Blocks, as shown in Table 3. We also employ the default reshaping (see Section B.1) on HHAR as target with 5-shot setting, and use the original source model input (miniImageNet) as reference, marked as Original. We examine the single-source baselines, including Transfer Learning (TL) (Jiang et al., 2022), Fine-tuning (FT) (Oh et al., 2022), MAML (Chen et al., 2019), DAPN (Zhao et al., 2021), and MetaSense (Gong et al., 2019) as the oracle baseline, shown in Table 1. To precisely track the MMC shift changes at each layer during fine-tuning,

---

[3]The support set and query set are assumed to be drawn from the same data distribution (Triantafillou et al., 2020).

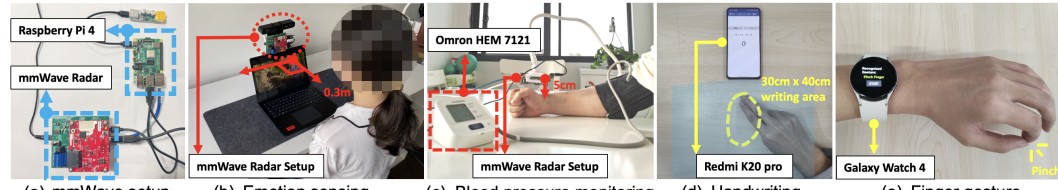

Figure 5: (a) Embedded mmWave radar testbed setup; (b)-(e) Built human sensing applications across different real-world settings.

## A.4 LAYER-WISE METRIC

To precisely track the MMC shift changes at each layer during fine-tuning, we essentially use the Silhouette score (S-score) (Shahapure & Nicholas, 2020) as the layer-wise metric to evaluate how well the features are being fine-tuned. The S-score is in the range from -1 to 1 and being better if closing to 1. To verify whether the S-score indicates the layer performance, we observe the changes in the S-score and accuracy at each layer using the TL (Jiang et al., 2022) and Original methods. Figure 6(a) also shows that S-score and accuracy by Original are positively correlated, converging in a layer-wise manner. TL's S-score also performs equivalently to its accuracy (*i.e.*, S-score drop along with accuracy drop) in detecting the negative transfer across different layers.

## A.5 IMPLEMENTATION

We implement `XTransfer` using PyTorch version 1.4.0. We also benchmark `XTransfer` with a full development cycle of cross-modality FSL tasks, including pre- and post-processing, training, and inference on commercial edge devices. Once pre-trained source models and few sensor data are processed in the cloud via `XTransfer`, compact models can be seamlessly converted to various DL platforms (*e.g.*, PyTorch Mobile, TFLite) for broader edge deployments. To enable reproducibility of the results, we fix all random seeds in the code, set the evaluation mode consistently when evaluating output models, and enable CUDA convolution determinism. To measure model resource overhead (*e.g.*, FLOPs and parameters), we use the standard PyTorch OpCounter. For regression tasks, class-based transformation (Shi et al., 2022) enables compatibility.

## A.6 DEVELOPED SENSING APPLICATIONS

To evaluate `XTransfer` on real-world human sensing targets, we essentially develop 4 human sensing applications, as shown in Figure 5. These include: (1) emotion recognition via facial expressions (*i.e.*, Emotion) and (2) non-contact blood pressure monitoring (*i.e.*, BP) based on an embedded mmWave radar setup, (3) handwriting tracking (*i.e.*, Writing) using ultrasound based on a smartphone, and (4) gesture recognition (*i.e.*, Gesture) using IMU and PPG sensors based on a smartwatch. The data collection is approved with ethics approval and involves 40 subjects. The corresponding target sensing datasets are shown in Table 2 with detailed specifications.

## A.7 HARDWARE SETUP AND COMPUTATION ENVIRONMENT

We evaluate `XTransfer` on real-world testbeds using 3 commercial edge devices, ranging from smartwatch to Raspberry Pi (Table 7). For example, Figure 5(a) presents an embedded mmWave radar setup which consists of a TI IWR1843Boost and a Raspberry Pi 4 using ROS Melodic on Ubuntu 18.04.4. In addition, we implement both smartwatch and smartphone testbeds using PyTorch Android version 1.10.0. We also employ servers equipped with NVIDIA GeForce RTX 3090 GPU to build our cloud environment.

## A.8 TRAINING DETAILS

In the Repair stage, we fine-tune the generative transfer module (see Section 4) for 100 episodes using SGD with a learning rate of 1e-2 and momentum 0.95. We apply a step-based learning rate scheduler with a step size of 20 and a decay factor of 0.5. We also use standard early stopping with a patience of 20. We retain the training parameters of the baselines in Table 1 as specified in their original implementations.

| Device | ROM | RAM | CPU | Battery | OS |
|---|---|---|---|---|---|
| Galaxy Watch 4 | 16GB | 1.5GB | Exynos W920 | 247mAh | Wear OS 3.5 |
| Raspberry Pi 4 | 64GB | 8GB | Broadcom BCM2711 | - | Ubuntu 18.04.4 |
| Redmi K20 Pro | 128GB | 8GB | Snapdragon 855 | 4000mAh | Android 11 |

Table 7: Edge devices specification.

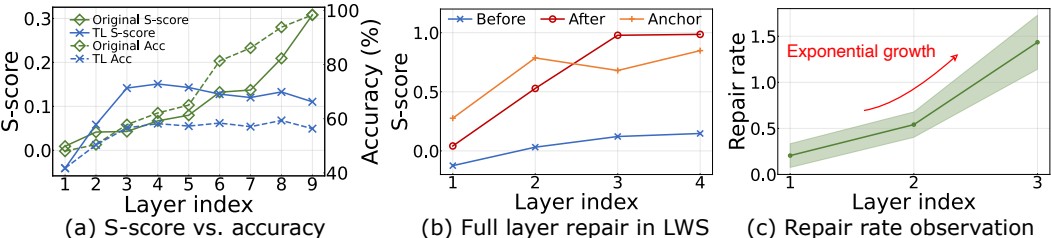

(a) S-score vs. accuracy  (b) Full layer repair in LWS  (c) Repair rate observation

Figure 6: Design insights. **(a)** shows layer-wise metric correlation. **(b)(c)** present efficient search insights into LWS control using multiple source models.

## B  TECHNICAL DETAILS

### B.1  DEFAULT RESHAPING

To align with source model input shape, we develop a *default reshaping* to transform sensor data shape. It uses bilinear interpolation (Mastyło, 2013) (*i.e.*, Resizer) to resize the height and width, and a fixed convolutional layer (*i.e.*, Fixed header) with a $1 \times 1$ kernel to match the input channels (Szegedy et al., 2014), placed between source models and sensor data.

### B.2  PCA-BASED LAYER CHANNEL REMOVAL

In the Removal stage, layer channel removal is designed to further enhance layer-wise performance while optimizing resource usage. This is achieved by removing unnecessary layer channels at each layer. Existing pruning techniques (Shen et al., 2022; Frankle & Carbin, 2019) typically rely on layer channel importance metrics, such as the L2 norm, calculated from MMC. However, due to MMC shift, these metrics may inaccurately assess layer channel importance, resulting in degraded layer performance. To address this, we propose using PCA-based layer channel importance, derived from the optimized component weights during the repair process, as a more reliable metric for layer channel removal. Our objective is to remove unnecessary layer channels $\widetilde{ch}$ while preserving the maximum S-score ($Score(\cdot)$), formulated as $\arg\max_{(\widetilde{ch} \in CH)} Score(PCA(MMC(ft_{ij}^r), \widetilde{ch}), Y_t)$, where $CH$ denotes the set of all layer channels, $r$ denotes the repaired latent features $ft_{ij}$, and $Y_t$ denotes the target labels.

### B.3  SETUPS FOR EFFICIENT SEARCH INSIGHTS

To gain design insights of efficient search for LWS control (see Section 5), we observe a complete process where all layers are repaired using the HHAR dataset and a Multi-ResNet18 model (*i.e.*, 5 source models with 45 L-unit), as shown in Table 3. Figure 6(b) shows the changes in the S-score across three states: after pairing anchors (*i.e.*, Anchor), before and after applying the SRR pipeline (*i.e.*, Before and After, respectively), in a layer-wise manner. The results show that the SRR pipeline successfully increases the S-score from Before state (*i.e.*, lower bound) to align with the Anchor state (*i.e.*, the objective). This indicates a strong correlation between the After state's S-score and both Before and Anchor values. These insights lead to the observation for designing the repair rate growth model.

### B.4  OBSERVATION FOR DESIGNING REPAIR RATE GROWTH MODEL

Our key idea is to estimate the repaired S-score (*i.e.*, repair value) of each layer *before* initiating SRR processing, allowing us to decide whether to bypass unnecessary repairs. To empirically examine this, we first define a repair rate function to capture this correlation as $rate_n = (After - Before)/Anchor = (V_{ij}^n - Score(Pro(ft_{ij}^n), Y_t))/Score(Pro(fs_{ij}^n), Y_s)$, where $Y_s$ denotes the

| Epoch (#) | #1 | #20 | #40 | #60 | #80 | #100 |
|---|---|---|---|---|---|---|
| With Alignment | 0.486 | 0.355 | 0.350 | 0.339 | 0.333 | 0.331 |
| Without Alignment | 0.476 | 0.406 | 0.393 | 0.383 | 0.378 | 0.387 |

Table 8: Feature space alignment performance across training epochs.

| PCA Variation | 3-shot | | 5-shot | | 10-shot | |
|---|---|---|---|---|---|---|
| | Accuracy (%) | Time (mins) | Accuracy (%) | Time (mins) | Accuracy (%) | Time (mins) |
| Standard PCA-#2 | 58.96 | 3.9 | 71.85 | 5.6 | 80.74 | 10.3 |
| Sparse PCA-#2 | 58.52 | 3.8 | 64.67 | 5.3 | 75.11 | 8.8 |

Table 9: Impact of PCA variations on accuracy and training time.

source labels, and we normalize $rate_n \in [0, 1]$. We then repeat the process multiple times to observe the changes in $rate_n$. Figure 6(c) shows that the repair rate demonstrates *exponential growth*. Thus, we design the repair rate growth model based on an exponential growth model (see Section 5.2).

### B.5 POST FINE-TUNING

After LWS control is completed, we add either a linear classifier or a regression layer (*e.g.*, a fully connected layer) at the end of the model to finalize its backbone. To strengthen layer-wise dependence, we perform a quick post fine-tuning process, updating only the connectors and fully connected layers.

## C ADDITIONAL EXPERIMENT RESULTS

### C.1 FEATURE SPACE ALIGNMENT PERFORMANCE

In the process of feature space alignment (Section 4.1), we use a standard Hungarian algorithm based on default distance metrics to select and pair classes. To verify its effectiveness, we keep using the setup of HHAR, the standard pre-trained ResNet18 on miniImageNet, and the standard PCA under 5-shot settings to examine how our repair loss performs with/without the feature space alignment, especially in the initial layer repairing. Repairing the initial layer is critical, as it often introduces a large, disorganized feature space and significantly affects the repair of subsequent layers. In Table 8, the results show that with the alignment, the repair loss decreases considerably faster by 1.87 times on average, compared to the case without alignment. This indicates that the design of feature space alignment notably contributes to the repairing process.

### C.2 IMPACT OF PCA VARIATIONS

Since our method uses the repair learning mechanism that relies on backpropagation (BP) through the reduced-orthogonal feature space, standard PCA can project data into a linear subspace and maintain a closed-form, differentiable structure, which enables smooth integration with BP in our end-to-end learning pipeline. Compared to advanced choices such as kernel PCA, we find that it introduces a non-linear mapping into a high-dimensional feature space using a kernel function, and its projections are generally not differentiable with respect to input features. This makes it incompatible with our pipeline. To test whether linear PCA variations affect the overall performance of XTransfer, we compare standard PCA against sparse PCA using the setup of HHAR based on the ResNet18 under 3 few-shot settings with a default search depth of 3. As shown in Table 9 (#2 denotes 2-component setting), sparse PCA reduces learning time slightly by 11.6% on average, but suffers from a notable accuracy drop by 4.4% on average. These results support the choice of standard PCA as a balanced and practical solution.

### C.3 IMPACT OF PCA COMPONENTS

We then evaluate the impact of PCA components on the model accuracy and time efficiency of XTransfer. Since the feature space with different dimensionalities adjusted by PCA components may impact our method performance, we test 3 different levels of PCA components, including PCA-#2 (top two components), PCA-80% (selected components covering 80% of the feature variance), and PCA-Max (maximum components available). We continue to use the same setup as above with

| PCA Components | 3-shot | | 5-shot | | 10-shot | |
|---|---|---|---|---|---|---|
| | Accuracy (%) | Time (mins) | Accuracy (%) | Time (mins) | Accuracy (%) | Time (mins) |
| PCA-#2 | 58.96 | 3.9 | 71.85 | 5.6 | 80.74 | 10.3 |
| PCA-80% | 63.61 | 5.2 | 74.72 | 7.6 | 80.00 | 12.9 |
| PCA-Max | 60.00 | 7.2 | 75.56 | 8.1 | 70.28 | 16.9 |

Table 10: Impact of PCA components on accuracy and training time.

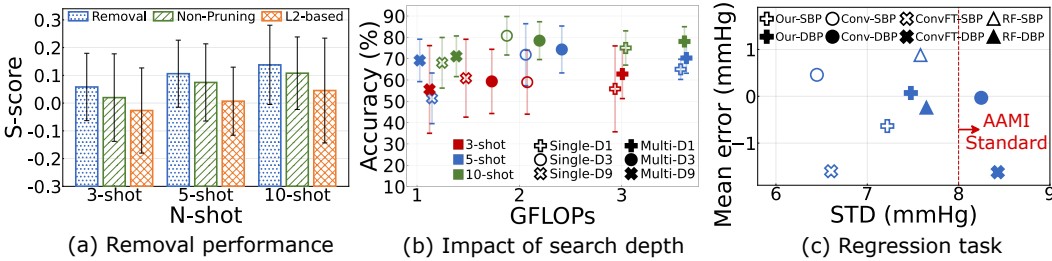

(a) Removal performance     (b) Impact of search depth     (c) Regression task

Figure 7: Ablation study evaluating the performance of components, search parameters, and applications.

the standard PCA. The results in Table 10 show that as the number of PCA components increases, the training time rises notably. In fact, PCA-Max results in an average accuracy drop of 4.5% compared to PCA-80%. By contrast, PCA-#2 achieves the best accuracy-to-time ratio across all n-shot settings, with competitive accuracy and the lowest training time, hence we use PCA-#2 as the default throughout the evaluation.

### C.4 REMOVAL PERFORMANCE

We next evaluate how layer channel removal enhances the average S-score of selected layers. We continue to use the same setup as above under 5-shot settings. We employ L2 norm-based (*i.e.*, L2-based) model pruning (Frankle & Carbin, 2019; Shen et al., 2022) as the baseline and compare it with the setting where layer channel removal is disabled (*i.e.*, Non-Pruning). Figure 7(a) demonstrates that the proposed layer channel removal outperforms both Non-Pruning and L2-based pruning across all settings, achieving an average S-score of 0.1 by selected layers. L2-based pruning achieves a lower average S-score than Non-Pruning, indicating that the layer channel importance may be inaccurately estimated, resulting in degraded performance. These results show that the proposed layer channel removal further improves layer performance by accurately removing unnecessary layer channels at each layer.

### C.5 IMPACT OF SEARCH DEPTH

In this experiment, we investigate how different search depths affect the accuracy and resource overhead (*e.g.*, FLOPs) of output models. Varying the search depth alters the search window sizes with the number of layer candidates, which can lead to different output model backbones and accuracy. To quantify these differences, we employ HHAR as target, ResNet18 on miniImageNet (Single) and Multi-ResNet18 (Multi) on 5 image datasets as sources (shown in Table 2) in 3-, 5- and 10-shot settings. We also utilize MetaSense in Table 1 as the oracle baseline. Both Single and Multi models have up to 9 L-unit, hence we set three different search depths as D1, D3, D9. Figure 7(b) shows that the output models of both Single-D1 and Multi-D1 result in high FLOPs, while that of both Single-D9 and Multi-D9 achieve low FLOPs. As search depth decreases, the search process becomes longer, and it triggers LWS control to make more decisions on recombining layers, which results in a suboptimal model backbone with high FLOPs. Although both Single-D3 and Multi-D3 do not produce the lowest FLOPs on average, they outperform others and achieve the highest average model accuracy of 70.52% and 70.69%, respectively, across the three settings. Therefore, we set D3 as the default search depth, as it provides the best balance between resource overhead and accuracy of the output models.

### C.6 IMPACT OF MODEL BACKBONES

The current implementation of XTransfer uses homogeneous pre-trained source models. Technically, the layer-wise repairing and recombination techniques impose no fundamental restrictions on

| Backbone | Accuracy (%) | FLOPs (G) |
|---|---|---|
| Multi-Conv4 | 65.56 | 0.01 |
| Multi-ResNet10 | 63.61 | 0.14 |
| Multi-ResNet18 | 78.44 | 2.19 |

Table 11: Performance of different source model backbones in terms of accuracy and FLOPs.

| Method | HHAR (mins) | | | WESAD (mins) | | | Gesture (mins) | | | Writing (mins) | | | Emotion (mins) | | | ChestX (mins) | | |
|---|---|---|---|---|---|---|---|---|---|---|---|---|---|---|---|---|---|---|
| | 3-shot | 5-shot | 10-shot | 3-shot | 5-shot | 10-shot | 3-shot | 5-shot | 10-shot | 3-shot | 5-shot | 10-shot | 3-shot | 5-shot | 10-shot | 3-shot | 5-shot | 10-shot |
| DAPN (Zhao et al., 2021) | 2.38 | 4.62 | 11.78 | 1.49 | 2.54 | 6.78 | 3.36 | 6.12 | 15.72 | 4.12 | 7.71 | 20.97 | 2.99 | 7.34 | 18.86 | 2.14 | 4.57 | 12.04 |
| MAML (Chen et al., 2019) | 124.28 | 125.66 | 183.20 | 71.71 | 77.07 | 115.87 | 118.79 | 129.79 | 191.95 | 165.06 | 129.27 | 194.36 | 94.81 | 98.40 | 153.20 | 66.20 | 72.17 | 106.97 |
| SemiCMT (Chen et al., 2024) | 3.02 | 8.27 | 10.23 | 1.82 | 2.48 | 5.63 | 4.81 | 9.42 | 12.83 | 10.10 | 13.92 | 18.75 | 3.39 | 7.32 | 11.48 | 2.84 | 5.28 | 8.33 |
| Our-Single | 3.98 | 5.57 | 7.95 | 2.12 | 2.99 | 4.33 | 5.64 | 7.80 | 11.24 | 7.94 | 10.54 | 14.69 | 5.58 | 7.43 | 9.88 | 3.65 | 4.75 | 6.84 |
| Our-Multi | 9.25 | 15.33 | 15.25 | 4.40 | 5.38 | 7.21 | 12.49 | 15.12 | 24.37 | 15.35 | 17.36 | 32.38 | 11.72 | 13.39 | 19.75 | 8.41 | 9.33 | 13.61 |

Table 12: Comparison in training time of each method.

the choice of model backbone and work with various architectural variants. To verify this, we evaluate the trade-off between accuracy and computational cost using different source backbones. We continue to use HHAR, and test Multi-Conv4, Multi-ResNet10, and Multi-ResNet18 in a 5-shot, multi-source setting shown in Table 3. The results in Table 11 demonstrate that the average accuracy declined, transitioning from Multi-Conv4 to Multi-ResNet10, and then increasing with Multi-ResNet18. Interestingly, Multi-Conv4, despite having fewer parameters, outperforms Multi-ResNet10, which may be attributed to differences in model quality or distributional divergence between source and target. Multi-ResNet18, though with higher FLOPs, achieves the best accuracy. It suggests that larger models trained on large-scale cross-modality datasets exhibit superior latent features, which notably contribute to such model transfer. Our techniques such as the connectors are designed to flexibly handle the layers with different shapes/structures, and can be extended to heterogeneous model backbones. In addition, our method focuses on latent feature alignment rather than specific feature extraction. It remains fully compatible with advanced source models (*e.g.*, with temporal encoders), which can be seamlessly integrated into our pipelines with minimal or no modifications.

## C.7 TRAINING EFFICIENCY

To test the practical training efficiency in the cloud, we further evaluate training time across the selected baselines. We use the same experimental setups as in Table 6, Section 6.3. For MAML (Chen et al., 2019) and SemiCMT (Chen et al., 2024), which require model re-training from pre-trained source models to adapt to the target modality classes, training time includes both re-training and fine-tuning. For DAPN (Zhao et al., 2021), it reflects model adaptation process on few target data. The results in Table 12 show that both Our-Single and Our-Multi achieve significant training time reductions compared to MAML. Our-Single also matches or exceeds DAPN and SemiCMT in training efficiency. Notably, even when using up to 5 source models, Our-Multi remains practical training efficiency, requiring only 2.05 times more training time on average compared to Our-Single across all datasets and shot settings. This further supports the practicability and scalability of our method.

## C.8 REGRESSION TASK PERFORMANCE

We also evaluate the effectiveness of `XTransfer` in a regression task using the BP dataset, which includes both systolic (SBP) and diastolic (DBP) sensor data shown in Table 2 collected by the blood pressure monitoring application. We employ the pre-trained ResNet18 on miniImageNet as the source model and use the 5-shot setting. To integrate this into the SRR pipeline, we transform the regression task by treating individual regression labels as distinct classes for anchor class pairing. For benchmarking, we use mean error (ME) and standard deviation (STD), following the Advancement of Medical Instrumentation (AAMI) standard, which accepts an ME within 5 mmHg and STD within 8 mmHg (Stergiou et al., 2018). We employ a random forest (RF) based method (Shi et al., 2023) as the oracle baseline. Besides, we train a Conv4 backbone in Table 3 (Conv) using LOOCV, and the fine-tuned Conv (ConvFT) as the baselines. Figure 7(c) shows that both `XTransfer` and the RF methods surpass the AAMI standard. In particular, `XTransfer` outperforms the baselines and RF in estimating both SBP (ME: -0.64mmHg, STD: 7.2mmHg) and DBP (ME: 0.07mmHg, STD: 7.47mmHg) tasks. These results demonstrate that `XTransfer` is highly efficient in handling regression tasks with few sensor data.

| Dataset | Type | FLOPs | Params | Watch | | Pi | | Phone | |
|---|---|---|---|---|---|---|---|---|---|
| | | | | Latency (ms) | Mem (MB) | Latency (ms) | Mem (MB) | Latency (ms) | Mem (MB) |
| HHAR | Single | 1.51G | 0.75M | 625.5 | 125.5 | 274.18 | 246.62 | 26.86 | 179.5 |
| | Multi | 1.46G | 2.16M | 634.5 | 140 | 249.26 | 247.83 | 28.34 | 209 |
| WESAD | Single | 1.19G | 0.74M | 604.5 | 120 | 240.05 | 244.64 | 25.77 | 168 |
| | Multi | 0.89G | 0.34M | 460.5 | 122 | 209.52 | 235.55 | 21.57 | 181 |
| Gesture | Single | 1.21G | 2.25M | 706.5 | 135 | 259.25 | 251.26 | 31.5 | 192 |
| | Multi | 1.45G | 2.09M | 692 | 137 | 268.79 | 253.7 | 30.96 | 190 |
| Writing | Single | 1.32G | 0.87M | 467.5 | 126 | 261.71 | 242.33 | 29.79 | 183.5 |
| | Multi | 1.28G | 0.73M | 513 | 124.5 | 285.2 | 247.01 | 34.67 | 185 |
| Emotion | Single | 1.51G | 0.49M | 767.5 | 130.5 | 390.36 | 271.02 | 41.1 | 171.5 |
| | Multi | 1.43G | 2.14M | 781 | 142 | 285.16 | 251.48 | 37.9 | 189.5 |
| ChestX | Single | 0.64G | 0.72M | 690.5 | 126 | 210.45 | 237.45 | 29.57 | 165 |
| | Multi | 1.06G | 0.7M | 670 | 130.5 | 261.14 | 236.96 | 30.14 | 183.5 |
| ResNet18 | - | 3.67G | 11.18M | 1060 | 189 | 703.52 | 303.75 | 89.8 | 311 |
| Multi-ResNet18 | - | 14.68G | 55.9M | 5816 | 356 | 4242.27 | 482.26 | 454.44 | 436 |

Table 13: On-device Performance.

| Alignment Method | 3-shot | | 5-shot | | 10-shot | |
|---|---|---|---|---|---|---|
| | Accuracy (%) | ATR | Accuracy (%) | ATR | Accuracy (%) | ATR |
| MMD Loss (Wang et al., 2023) | 55.78 | 1.63 | 65.89 | 2.01 | 68.52 | 1.83 |
| Repair Loss | 65.11 | 2.33 | 69.33 | 2.45 | 72.07 | 2.29 |
| PCA + Repair Loss | 58.96 | 1.42 | 71.85 | 1.56 | 80.74 | 2.09 |

Table 14: Comparison of non-linear feature alignment methods across n-shot settings.

## C.9 ON-DEVICE PERFORMANCE

We now evaluate the on-device performance of deploying XTransfer's output models on 3 commercial edge devices (Table 7), focusing on resource overhead metrics such as FLOPs, parameters, latency per call, and peak memory footprint. For each target dataset, we test both types of output models created by our single-source (Single) and multi-source (Multi) methods. We also use ResNet18 and Multi-ResNet18 (Table 3) as baselines. As shown in Table 13, all output models achieve significant reductions in model resource overhead, *e.g.*, FLOPs is reduced by 2.4 to 5.7× for Single models and by 10.1 to 16.5× for Multi models, compared to the baselines. These recombined models also demonstrate substantial improvement in on-device performance. Specifically, in the Single method, latency is reduced by 1.4 to 4.1 × and peak memory footprint decreases by 1.8 to 3.9×. In the Multi method, the reductions are even more pronounced, with latency dropping by 7.4 to 21× and memory footprint by 2.2 to 8.4×, compared to the baselines. These results prove that XTransfer successfully achieves streamlined edge deployment.

## C.10 NON-LINEAR FEATURE ALIGNMENT PERFORMANCE

We essentially use PCA to construct a fixed, orthogonal anchor feature space in which alignment is measured and the anchor-based repair loss is computed at each layer across modalities. The proposed connector (*i.e.*, repair generator) dynamically maps target-modality features into this anchor space, enabling the non-linear feature alignment. To examine the impact of using a linear PCA anchor space and to compare the alignment performance, we evaluate our repair loss with and without PCA against a Maximum Mean Discrepancy (MMD) loss baseline (Wang et al., 2023). To set up, we keep using the standard ResNet18 on miniImageNet (Table 3) as sources on HHAR as target (Table 2) in 3-, 5- and 10-shot settings with all configurations identical except for the alignment loss. As shown in Table 14, our repair loss with PCA anchor space achieves the highest accuracy of 70.5% and the lowest ATR of 1.69 on average across all n-shot settings. Compared to our repair loss without PCA, MMD loss performs a lower accuracy on average. This indicates that MMD loss is limited to aligning the latent feature distribution when the feature space is highly complex due to strong cross-modality discrepancies and only few target data. In addition, adversarial alignment methods such as DAPN (Zhao et al., 2021) and MDDA (Zhao et al., 2020) also face similar difficulties under these conditions, as shown in Tables 4 and 5. Moreover, the PCA anchor space provides a more stable basis for alignment and consistently enhances our repair loss performance. As a result, these findings further validate the effectiveness of our method in cross-modality FSL settings.

| Epoch (#) | #10 | #20 | #30 | #40 | #50 | #60 | #70 | #80 | #90 | #100 |
|---|---|---|---|---|---|---|---|---|---|---|
| **MMD Loss** (Wang et al., 2023) | 6.46% | 10.0% | 11.63% | 12.81% | 13.5% | 14.3% | 14.66% | 14.98% | 15.27% | 15.41% |
| **Repair Loss** | 0.03% | 1.67% | 2.66% | 4.93% | 5.26% | 7.16% | 8.06% | 10.4% | 14.66% | 16.14% |
| **PCA + Repair Loss** | 5.12% | 13.45% | 16.54% | 19.23% | 20.55% | 21.3% | 25.96% | 26.21% | 27.6% | 28.87% |

Table 15: Comparison of layer-wise convergence rates across alignment methods.

## C.11 LAYER-WISE CONVERGENCE COMPARISON

To further examine layer-wise convergence across the above alignment methods, we observe the feature alignment process in the initial layer. To normalize heterogeneous losses, we follow the strategy in (Li et al., 2020) and compute the loss descent (LD) relative to the initial loss, enabling a fair comparison of convergence rates across different loss functions (*i.e.*, how much each loss decreases from its starting point), defined as $LD(t) = (L(0) - L(t)) / L(0) \times 100\%$, where $t$ denotes the epoch index. All experimental settings remain the same as in the previous setup. As shown in Table 15, our repair loss with the PCA anchor achieves the largest loss descent of 28.87%, which is 1.79 and 1.87 times faster than the repair loss without PCA and the MMD loss, respectively. Interestingly, MMD loss presents a large initial drop but quickly plateaus, while our repair loss starts more gradually and accelerates over training. In contrast, the PCA-based repair maintains a steady and consistent descent across epochs, highlighting the benefits of aligning features in a fixed, orthogonal anchor space. These findings further support that our method provides both effective feature alignment and practical training efficiency, consistent with the efficiency study in Section C.7.

## D THE USE OF LARGE LANGUAGE MODELS (LLMS)

The LLM is used for writing, editing, or formatting purposes and does not impact the core methodology, scientific rigorousness, or originality of the research. We also use GPT2 as a baseline in our experimental evaluation to compare against our proposed model transfer method. The core method development in this work does not involve LLMs as components.

