# OpenReview forum: "XTransfer: Modality-Agnostic Few-Shot Model Transfer for Human Sensing at the Edge"
_ICLR.cc/2026/Conference — Submitted to ICLR 2026_

### Official Review · Reviewer_F1Vc · 2025-10-31

**Soundness:** 2
**Presentation:** 2
**Contribution:** 2
**Rating:** 4
**Confidence:** 4

**Summary:**

This paper introduces a framework for modality-agnostic, few-shot model transfer tailored for human sensing on edge devices. The core contributions are a splice repair removal pipeline that mitigates modality shift by aligning latent feature distributions of target sensor data with pre-trained source models using an anchor-based loss in a reduced PCA space, and a layer wise search mechanism that efficiently searches and recombines useful layers from single or multiple source models to construct a compact, high-performance target model.

**Strengths:**

Tackles a practical problem at the intersection of few-shot learning, cross-modal transfer, and edge AI. The proposed method's ambition is to leverage readily available pre-trained models from vastly different modalities (e.g., image, text) for specialized sensing tasks. The quality of experimental evaluation is good.

**Weaknesses:**

The method's reliance on mean magnitude of channels and its s-score as the primary metric for guiding layer repairing and selection feels under-justified. While it  is presented as a lightweight metric, its suitability for capturing feature discriminability across drastically different modalities (e.g., vision to IMU) is not intuitively clear, and a more thorough justification or comparison against other feature distribution metrics (e.g., MMD) would strengthen this core design choice.

The proposed approach is also quite complex, involving multiple components, stages, and hyperparameters (e.g., PCA dimensionality, search parameters), which could pose challenges for reproducibility and practical implementation.

Finally, LWS could face scalability issues as the number and depth of source models increase and the robustness of the proposed $rate^{est}$ model for the pre-search check is not fully explored, especially in highly dissimilar transfer settings.

**Questions:**

The "Model Repairing" component centers on aligning feature spaces to minimize MMC shift. Could you elaborate on the intuition for using a channel-magnitude metric like MMC for cross-modality transfer, where feature representations are fundamentally different? Have you experimented with alternative feature alignment techniques, such as adversarial alignment or MMD, and how do they compare?

The LWS recombines layers sequentially from source models. How does this mechanism handle non-sequential architectural elements, such as the skip connections in ResNet architectures? Are these connections discarded, or does your method have a way to preserve or reconstruct them in the final compact model?

The pre-search check's efficiency depends on an exponential growth model for the repair rate. How was this specific model form chosen, and how robust is the search process if the actual repair rate for a given source-target pair deviates a lot from assumed exponential trend?

Table 4 shows that in the challenging 3-shot setting, XTransfer's accuracy is slightly below the oracle baseline on some tasks. Does this point to a fundamental limit on the minimum data required for stable alignment, and could this gap be closed by integrating few-shot data augmentation techniques into the SRR pipeline?

---

> ### Author Response · Authors · 2025-11-23
> **Response to Reviewer F1Vc (1/3)**
>
> We appreciate your insightful review and comments. We are glad that the reviewer acknowledges the ambition of the proposed method and "**the quality of experimental evaluation is good**". Based on the comments, we clarify some technical designs and address your concerns as below. In addition, we have submitted a revised manuscript with extended page and appendix in which we add all the suggested analytics, discussion, and comparison in yellow color.
>
>
> **Q1. The method's reliance on mean magnitude of channels and its s-score as the primary metric for guiding layer repairing and selection feels under-justified. a more thorough justification or comparison against other feature distribution metrics (e.g., MMD) would strengthen this core design choice. Could you elaborate on the intuition for using a channel-magnitude metric like MMC for cross-modality transfer, where feature representations are fundamentally different? Have you experimented with alternative feature alignment techniques, such as adversarial alignment or MMD, and how do they compare?**
>
> **Answer**: Thanks for the insightful comments. We adopt the MMC as a statistical feature similarity metric because it provides a lightweight yet effective measure of layer-wise feature channel consistency. MMC has been proven in prior model compression studies to correlate well with representational quality. Our method is compatible with alternative metrics as long as the correlation between layer-wise metrics and model accuracy holds, and MMC can be readily replaced with any qualified substitute. MMD is a distribution-level distance metric and can serve as a training loss for distribution alignment, so MMD can work jointly with MMC in our settings.
>
> To further strengthen the design choice, we have added the below ablation study in Appendix C.10 (line 1225) to examine the impact of using a linear PCA anchor space and to compare the alignment performance against the mentioned non-linear feature alignment methods.
>
> |                |    **3-shot**    |                 |    **5-shot**    |                 |   **10-shot**    |                   |
> |----------------|:----------------:|:---------------:|:----------------:|:---------------:|:----------------:|:-----------------:|
> | **Aligment Method** | **Accuracy (\%)** | **ATR** | **Accuracy (\%)** | **ATR** | **Accuracy (\%)** | **ATR** |
> | MMD Loss |     55.78       |       1.63      |      65.89       |       2.01      |      68.52       |       1.83        |
> | Repair Loss |      65.11      |       2.33       |      69.33     |       2.45      |      72.07      |      2.29       |
> | PCA + Repair Loss  |      58.96       |       1.42       |      71.85     |       1.56       |      80.74      |      2.09        |
>
> We evaluate our repair loss with and without PCA against a Maximum Mean Discrepancy (MMD) loss baseline [1]. To set up, we keep using the standard ResNet18 on miniImageNet as sources on HHAR as target in 3-, 5- and 10-shot settings with all configurations identical except for the alignment loss. As shown in above Table, our repair loss with PCA anchor space achieves the highest accuracy of 70.5\% and the lowest ATR of 1.69 on average across all n-shot settings. Compared to our repair loss without PCA, MMD loss performs a lower accuracy on average. This indicates that MMD loss is limited to aligning the latent feature distribution when the feature space is highly complex due to strong cross-modality discrepancies and only few target data. In addition, adversarial alignment methods such as DAPN [2] and MDDA [3] in our baselines also face similar difficulties under these conditions, as shown in Tables 4 (line 494) and 5 (line 503) in the main manuscript. Moreover, the PCA anchor space provides a more stable basis for alignment and consistently enhances our repair loss performance.
>
>
> **Q2. Pipeline is quite complex to have challenges for reproducibility and practical implementation.**
>
> **Answer**: As mentioned in the Reproducibility Statement (line 555), we have fully implemented XTransfer and attached in the Supplementary Material. We report the specifications of our computational environment in Appendix A, including hardware setups (CPUs, GPUs, and RAM), memory consumption, and software dependencies. These details are provided to support transparency and reproducibility across different computational platforms. We are also committed to release our source code in a public GitHub repository, accompanied by detailed instructions to guide researchers in reproducing the experimental results.

---

> ### Author Response · Authors · 2025-11-23
> **Response to Reviewer F1Vc (2/3)**
>
> **Q3. LWS could face scalability issues as the number and depth of source models increase and the robustness of the proposed model for the pre-search check is not fully explored, especially in highly dissimilar transfer settings. The pre-search check's efficiency depends on an exponential growth model for the repair rate. How was this specific model form chosen, and how robust is the search process if the actual repair rate for a given source-target pair deviates a lot from assumed exponential trend?**
>
>
> **Answer**: Thanks for the valuable questions. As discussed in Appendix B.5 (now moved to Section 5.2, line 363 for clearer method presentation), we proposed the dynamic search range mechanism to stabilize the pre-search check. The pre-search check uses the repair model to estimate repair values, rank layers, and select the top-1 or top-2 layers while skipping the rest. However, when these estimated values are inaccurate or drift over layers, the search becomes unstable. To address this, we design dynamic range control mechanism. Instead of fixing the search range to only select the top-1 or top-2 layers, we dynamically expand or shrink the range of layers that proceed to actual repairing. A larger range repairs more layers and improves accuracy, while a smaller range bypasses unnecessary repairs and reduces computation, enabling an explicit accuracy–efficiency trade-off. Furthermore, the repair rate growth model is motivated by the empirical observation in Appendix B.4 (line 1023), where the actual repair rate follows an exponential growth pattern.
>
> To further examine the search stability, we have extended the efficient search performance study in Figure 4(b) (line 394) and Section 6.2 (line 437) with new results below.
>
> |                |    **3-shot**    |                 |    **5-shot**    |                 |   **10-shot**    |                   |
> |----------------|:----------------:|:---------------:|:----------------:|:---------------:|:----------------:|:-----------------:|
> | **Search Method** | **Accuracy (\%)** | **Time (mins)** | **Accuracy (\%)** | **Time (mins)** | **Accuracy (\%)** | **Time (mins)** |
> | Single |     58.96 ± 15.00     |       3.98      |      71.85 ± 14.53      |       5.57      |      80.74 ± 9.03      |       10.33        |
> | Multi |      51.67 ± 16.79     |       19.42       |      69.26 ± 10.12    |       30.82      |      77.41 ± 6.44     |      55.81       |
> | Multi-Pre  |      54.68 ± 18.05      |       3.90       |      64.05 ± 12.77    |       4.33       |      69.52 ± 9.82     |      5.39        |
> | Multi-Efficient  |      59.33 ± 15.06      |       9.25       |      74.30 ± 11.00     |       10.55       |      78.44 ± 8.87      |      14.03        |
>
> We compare different configurations: ResNet18 (Single) and Multi-ResNet18 (Multi) without efficient search enabled, Multi-ResNet18 (Multi-Pre) with only pre-search check using top-1 layer selection enabled, and Multi-ResNet18 (Multi-Efficient) with both pre-search check and dynamic search range enabled. As shown in above Table, Multi-Pre achieves a lowest search time on average across all n-shot settings, but it leads to degraded accuracy by 3.36\% and 7.94\% on average, respectively, compared to Multi and Multi-Efficient. It also fails to reach oracle-level accuracy and presents the highest standard deviation on average. The results indicate that relying solely on the pre-search check introduces search stability issues due to inaccurate or drifted repair value estimation. In contrast, Multi-Efficient provides a strong balance between accuracy and efficiency. It reduces search time by 2.1 to 4 times compared to Multi in 5- and 10-shot settings, while outperforming the oracle baseline. While Multi-Efficient takes a longer search time than both Single and Multi-Pre, it successfully achieves superior accuracy and search stability, highlighting the benefits of using multiple source models and dynamic search range mechanism. Notably, even the search space is expanded 5 times more compared to Single, Multi-Efficient requires only 2.1 times more search time, indicating strong scalability enabled by the efficient search. The results also reveal that both Multi-Efficient and Multi achieve lower standard deviation on average compared to Single, suggesting that using multiple source models offers commendable search stability. In short, the results prove that the proposed efficient search significantly accelerates the layer search process while improving accuracy and stability.

---

> ### Author Response · Authors · 2025-11-23
> **Response to Reviewer F1Vc (3/3)**
>
> **Q4. The LWS recombines layers sequentially from source models. How does this mechanism handle non-sequential architectural elements, such as the skip connections in ResNet architectures? Are these connections discarded, or does your method have a way to preserve or reconstruct them in the final compact model?**
>
> **Answer**: Thanks for the thoughtful questions. As clarified in Section 3 (line 203), XTransfer uses either a single layer or a dependent layer block (i.e., an L-Block, where a group of layers are not structurally separable) as the default segmentation unit. For non-sequential architectural elements such as the skip connections in ResNet, we simply keep the skip connections within each layer block and select among the layer blocks, allowing XTransfer to operate consistently across different model architectures.
>
>
>
> **Q5. Table 4 shows that in the challenging 3-shot setting, XTransfer's accuracy is slightly below the oracle baseline on some tasks. Does this point to a fundamental limit on the minimum data required for stable alignment, and could this gap be closed by integrating few-shot data augmentation techniques into the SRR pipeline?**
>
> **Answer**: Thanks for the insightful questions. We have added the discussion of failure cases and extended our sensitivity analysis under 2-shot settings in Figure 4(c) (line 394), Section 6.2 (line 461) and 6.3 (line 515) with new results below.
>
> |                |    **2-shot**    |                 |    **3-shot**    |                 |   **5-shot**    |                   |   **10-shot**    |                  |
> |----------------|:----------------:|:---------------:|:----------------:|:---------------:|:----------------:|:-----------------:|:----------------:|:-----------------:|
> | **Source Modality** | **Accuracy (\%)** | **GFLOPs** | **Accuracy (\%)** | **GFLOPs** | **Accuracy (\%)** | **GFLOPs** | **Accuracy (\%)** | **GFLOPs** |
> | Text |     59.33       |       0.14      |      64.29       |       0.12      |      70.59       |       0.17       |      73.93       |       0.23        |
> | Audio |      59.19      |       2.01       |      60.52     |       1.75      |      63.41      |      2.04       |      68.89       |       1.73        |
> | Sensing  |      61.43      |       0.15       |      65.28     |        0.11       |      67.22     |      0.18        |      68.61       |       0.22        |
> | Image  |      56.67       |       1.81       |      58.96     |       2.08       |      71.85      |      2.06        |      80.74       |       1.88        |
> | Oracle  |      61.33       |       -      |      63.19    |       -       |      68.96      |      -        |      75.19       |       -        |
>
> The results show that Image achieves lower average accuracy in 2- and 3-shot settings, suggesting that extremely low target data may not fully represent the test data distribution, leading to less stable alignment when using Image source. In contrast, both Text and Sensing result in higher average accuracy and reach the oracle, indicating that performance varies with source model quality and semantic relevance to the target. These results also highlight that XTransfer can flexibly reuse diverse source models across modalities to improve ultra low-shot performance. In addition, integrating few-shot data augmentation techniques could further help close the gap, and such methods are orthogonal and can be jointly integrated with XTransfer.
>
>
> **Reference:**
>
> [1] Wang, W., Li, H., Ding, Z., Nie, F., Chen, J., Dong, X., and Wang, Z. (2023). Rethinking maximum mean discrepancy for visual domain adaptation. IEEE Transactions on Neural Networks and Learning Systems.
>
> [2] Zhao, A., Ding, M., Lu, Z., Xiang, T., Niu, Y., Guan, J., and Wen, J.-R. (2021). Domain-adaptive few-shot learning. In Proceedings of WACV 2021.
>
> [3] Zhao, S., Wang, G., Zhang, S., Gu, Y., Li, Y., Song, Z., Xu, P., Hu, R., Chai, H., and Keutzer, K. (2020). Multi-source distilling domain adaptation. In Proceedings of AAAI 2020.

---

> ### Author Response · Authors · 2025-11-27
> **Looking forward to your reply**
>
> Dear reviewer F1Vc,
>
> As the rebuttal deadline is approaching, could you please have a look at our response? We are looking forward to your reply!
>
> Feel free to let us know if you have any other concerns. Thanks for your time and effort!
>
> Best Regards,
>
> Authors of Submission 12243

---

### Official Review · Reviewer_yVS9 · 2025-10-31

**Soundness:** 3
**Presentation:** 4
**Contribution:** 3
**Rating:** 6
**Confidence:** 4

**Summary:**

This paper introduces a modality-agnostic few-shot transfer learning framework tailored for resource-constrained human-sensing applications on edge devices.
The method leverages pre-trained models as sources and combines layer repairing to mitigate modality shift with layer-wise recombination to select only beneficial layers, thereby producing compact, efficient models.
The authors evaluated the proposed method across several sensing datasets and showed state-of-the-art accuracy while reducing sensor data requirements, training cost, and deployment resource overhead.

**Strengths:**

- The paper identifies a timely and well-motivated challenge in human-sensing systems and tackles few-shot cross-modality transfer on resource-constrained edge devices.
- The proposed XTransfer framework integrates a structured SRR pipeline for modality repair with a principled layer-wise recombination strategy. The design addresses both representation alignment and parameter efficiency, demonstrating a thoughtful mechanism for reusing heterogeneous pre-trained models.
- The study evaluates the approach across multiple sensing modalities, diverse benchmarks, and real edge-device settings. Results consistently show improvements in accuracy-resource trade-offs and training efficiency, providing convincing evidence of the method's scalability and practicality for deployment.

**Weaknesses:**

- The paper would benefit from a discussion of failure cases, sensitivity to noisy or highly heterogeneous sensor data, and robustness under severe domain shifts
- The SRR and layer-wise search procedures introduce methodological complexity, and the paper does not fully quantify the tuning burden, search overhead under diverse hardware constraints, or potential stability issues when scaling to larger sets of heterogeneous source models.
- The evaluation focuses primarily on cross-modality human-sensing tasks, and comparison to broader transfer paradigms (e.g., recent foundation-model or prompt-based adaptation techniques) is limited.

**Questions:**

See weaknesses.

---

> ### Author Response · Authors · 2025-11-23
> **Response to Reviewer yVS9 (1/3)**
>
> We sincerely thank you for the positive assessment. We are grateful for your insightful questions and feedback, and pleased that you found "**demonstrating a thoughtful mechanism and providing convincing evidence of the method's scalability and practicality for deployment**". Based on the suggestions, we address the detailed questions as below. In addition, we have submitted a revised manuscript with extended page and appendix in which we add all the suggested analytics, discussion, and comparison in yellow color.
>
> **Q1. The paper would benefit from a discussion of failure cases, sensitivity to noisy or highly heterogeneous sensor data, and robustness under severe domain shifts.**
>
> **Answer**: Thanks for the valuable suggestions. We have added the discussion of failure cases and extended our sensitivity analysis under 2-shot settings in Figure 4(c) (line 394), Section 6.2 (line 461) and 6.3 (line 515) with new results below.
>
> |                |    **2-shot**    |                 |    **3-shot**    |                 |   **5-shot**    |                   |   **10-shot**    |                  |
> |----------------|:----------------:|:---------------:|:----------------:|:---------------:|:----------------:|:-----------------:|:----------------:|:-----------------:|
> | **Source Modality** | **Accuracy (\%)** | **GFLOPs** | **Accuracy (\%)** | **GFLOPs** | **Accuracy (\%)** | **GFLOPs** | **Accuracy (\%)** | **GFLOPs** |
> | Text |     59.33       |       0.14      |      64.29       |       0.12      |      70.59       |       0.17       |      73.93       |       0.23        |
> | Audio |      59.19      |       2.01       |      60.52     |       1.75      |      63.41      |      2.04       |      68.89       |       1.73        |
> | Sensing  |      61.43      |       0.15       |      65.28     |        0.11       |      67.22     |      0.18        |      68.61       |       0.22        |
> | Image  |      56.67       |       1.81       |      58.96     |       2.08       |      71.85      |      2.06        |      80.74       |       1.88        |
> | Oracle  |      61.33       |       -      |      63.19    |       -       |      68.96      |      -        |      75.19       |       -        |
>
> The results show that Image achieves lower average accuracy in 2- and 3-shot settings, suggesting that extremely low target data may not fully represent the test data distribution, leading to less stable alignment when using Image source. In contrast, both Text and Sensing result in higher average accuracy and reach the oracle, indicating that performance varies with source model quality and semantic relevance to the target. These results also highlight that XTransfer can flexibly reuse diverse source models across modalities to improve ultra low-shot performance.
>
>
>
> **Q2. The SRR and layer-wise search procedures introduce methodological complexity, and the paper does not fully quantify the tuning burden, search overhead under diverse hardware constraints, or potential stability issues when scaling to larger sets of heterogeneous source models.**
>
> **Answer**: Thanks for the insightful comments. To further quantify the SRR performance and the search stability, we have added the below ablation studies with new results in the revised manuscript:
>
> - We have added the below ablation study in Appendix C.10 (line 1225) to examine the impact of using a linear PCA anchor space and to compare the alignment performance against the mentioned non-linear feature alignment methods by Reviewer F1Vc.
>
> |                |    **3-shot**    |                 |    **5-shot**    |                 |   **10-shot**    |                   |
> |----------------|:----------------:|:---------------:|:----------------:|:---------------:|:----------------:|:-----------------:|
> | **Aligment Method** | **Accuracy (\%)** | **ATR** | **Accuracy (\%)** | **ATR** | **Accuracy (\%)** | **ATR** |
> | MMD Loss |     55.78       |       1.63      |      65.89       |       2.01      |      68.52       |       1.83        |
> | Repair Loss |      65.11      |       2.33       |      69.33     |       2.45      |      72.07      |      2.29       |
> | PCA + Repair Loss  |      58.96       |       1.42       |      71.85     |       1.56       |      80.74      |      2.09        |
>
> We evaluate our repair loss with and without PCA against a Maximum Mean Discrepancy (MMD) loss baseline [1]. To set up, we keep using the standard ResNet18 on miniImageNet as sources on HHAR as target in 3-, 5- and 10-shot settings with all configurations identical except for the alignment loss. As shown in above Table, our repair loss with PCA anchor space achieves the highest accuracy of 70.5\% and the lowest ATR of 1.69 on average across all n-shot settings. Compared to our repair loss without PCA, MMD loss performs a lower accuracy on average.

---

> ### Author Response · Authors · 2025-11-23
> **Response to Reviewer yVS9 (2/3)**
>
> **(Here is continued with above 1/3)**
>
> This indicates that MMD loss is limited to aligning the latent feature distribution when the feature space is highly complex due to strong cross-modality discrepancies and only few target data. In addition, adversarial alignment methods such as DAPN [2] and MDDA [3] in our baselines also face similar difficulties under these conditions, as shown in Tables 4 (line 494) and 5 (line 503) in the main manuscript. Moreover, the PCA anchor space provides a more stable basis for alignment and consistently enhances our repair loss performance.
>
> - We have added the below ablation study in Appendix C.11 (line 1249) to compare our repair loss with other non-linear feature alignment loss methods, presenting a layer-wise convergence rate comparison.
>
> | **Epoch(#)** | **#10** | **#20** | **#30** | **#40** | **#50** | **#60** | **#70** | **80** | **#90** | **#100** |
> |--------------|:------:|:-------:|:-------:|:-------:|:-------:|:--------:|:-------:|:------:|:-------:|:--------:|
> | MMD Loss [1]    | 6.46\%  |  10.0\%  |  11.63\%  |  12.81\%  |  13.5\%  |  14.3\%   |  14.66\%  | 14.98\%  |  15.27\%  |  15.41\%   |
> | Repair Loss    | 0.03\% | 1.67\%  | 2.66\%   | 4.93\%  | 5.26\%  | 7.16\%  | 8.06\%  | 10.4\%  | 14.66\% | 16.14\% |
> | PCA + Repair Loss | 5.12\% | 13.45\% | 16.54\% | 19.23\% | 20.55\% | 21.3\% | 25.96\% | 26.21\% | 27.6\% | 28.87\% |
>
> We observe the feature alignment process in the initial layer which often introduces a large, disorganized feature space and significantly affects the repair of subsequent layers. To normalize heterogeneous losses, we follow the strategy in [4] and compute the loss descent (LD) relative to the initial loss, enabling a fair comparison of convergence rates across different loss functions (i.e., how much each loss decreases from its starting point). All experimental settings remain the same as in the previous setup. As shown in above Table, our repair loss with the PCA anchor achieves the largest loss descent of 28.87\%, which is 1.79 and 1.87 times faster than the repair loss without PCA and the MMD loss, respectively. Interestingly, MMD loss presents a large initial drop but quickly plateaus, while our repair loss starts more gradually and accelerates over training. In contrast, the PCA-based repair maintains a steady and consistent descent across epochs, highlighting the benefits of aligning features in a fixed, orthogonal anchor space. These findings further support that our method provides both effective feature alignment and practical training efficiency, consistent with the efficiency study in Appendix C.7 (line 1163).
>
> - We have extended the efficient search performance study to further examine the search stability in Figure 4(b) (line 394) and Section 6.2 (line 437) with new results below.
>
> |                |    **3-shot**    |                 |    **5-shot**    |                 |   **10-shot**    |                   |
> |----------------|:----------------:|:---------------:|:----------------:|:---------------:|:----------------:|:-----------------:|
> | **Search Method** | **Accuracy (\%)** | **Time (mins)** | **Accuracy (\%)** | **Time (mins)** | **Accuracy (\%)** | **Time (mins)** |
> | Single |     58.96 ± 15.00     |       3.98      |      71.85 ± 14.53      |       5.57      |      80.74 ± 9.03      |       10.33        |
> | Multi |      51.67 ± 16.79     |       19.42       |      69.26 ± 10.12    |       30.82      |      77.41 ± 6.44     |      55.81       |
> | Multi-Pre  |      54.68 ± 18.05      |       3.90       |      64.05 ± 12.77    |       4.33       |      69.52 ± 9.82     |      5.39        |
> | Multi-Efficient  |      59.33 ± 15.06      |       9.25       |      74.30 ± 11.00     |       10.55       |      78.44 ± 8.87      |      14.03        |
>
> We compare different configurations: ResNet18 (Single) and Multi-ResNet18 (Multi) without efficient search enabled, Multi-ResNet18 (Multi-Pre) with only pre-search check using top-1 layer selection enabled, and Multi-ResNet18 (Multi-Efficient) with both pre-search check and dynamic search range enabled. As shown in above Table, Multi-Pre achieves a lowest search time on average across all n-shot settings, but it leads to degraded accuracy by 3.36\% and 7.94\% on average, respectively, compared to Multi and Multi-Efficient. It also fails to reach oracle-level accuracy and presents the highest standard deviation on average. The results indicate that relying solely on the pre-search check introduces search stability issues due to inaccurate or drifted repair value estimation. In contrast, Multi-Efficient provides a strong balance between accuracy and efficiency. It reduces search time by 2.1 to 4 times compared to Multi in 5- and 10-shot settings, while outperforming the oracle baseline.

---

> ### Author Response · Authors · 2025-11-23
> **Response to Reviewer yVS9 (3/3)**
>
> **(Here is continued with above 2/3)**
>
> While Multi-Efficient takes a longer search time than both Single and Multi-Pre, it successfully achieves superior accuracy and search stability, highlighting the benefits of using multiple source models and dynamic search range mechanism. Notably, even the search space is expanded 5 times more compared to Single, Multi-Efficient requires only 2.1 times more search time, indicating strong scalability enabled by the efficient search. The results also reveal that both Multi-Efficient and Multi achieve lower standard deviation on average compared to Single, suggesting that using multiple source models offers commendable search stability. In short, the results prove that the proposed efficient search significantly accelerates the layer search process while improving accuracy and stability.
>
> For search overhead, the efficient search, including both of the pre-search check based on repair rate growth model (Section 5.2, line 349) and the dynamic search range mechanism (moved from Appendix B.5 to Section 5.2, line 363), is designed to be lightweight. The LWS value function also employs the efficient S-score metric, ensuring that LWS introduces minimal overhead, compared to the SRR repair process. We reported training resource usage in Table 6 (line 510) and overall time efficiency in Appendix Table 12 (line 1145). The results show no significant increase in GPU utilization. Even when using multiple source models, the total training time increases by only 2.05× on average compared with using a single source model.
>
>
>
> **Q3. The evaluation focuses primarily on cross-modality human-sensing tasks, and comparison to broader transfer paradigms (e.g., recent foundation-model or prompt-based adaptation techniques) is limited.**
>
> **Answer**: Thanks for the thoughtful comments. While this paper focuses on human sensing tasks, XTransfer supports the full cycle of cross-modality FSL tasks, including training, inference and benchmarking, and can be extended to other target domains facing the challenges of data scarcity and resource constraints. Regarding broader transfer paradigms such as foundation-model methods, they are not designed for modality-agnostic few-shot model transfer, but we plan to extend the comparisons in future work. We also revised the Related Work section (line 146) to add the discussions with the foundation models (FM) methods [5, 6] and the prompt-based FM adaptation [7].
>
> **Reference:**
>
> [1] Wang, W., Li, H., Ding, Z., Nie, F., Chen, J., Dong, X., and Wang, Z. (2023). Rethinking maximum mean discrepancy for visual domain adaptation. IEEE Transactions on Neural Networks and Learning Systems.
>
> [2] Zhao, A., Ding, M., Lu, Z., Xiang, T., Niu, Y., Guan, J., and Wen, J.-R. (2021). Domain-adaptive few-shot learning. In Proceedings of WACV 2021.
>
> [3] Zhao, S., Wang, G., Zhang, S., Gu, Y., Li, Y., Song, Z., Xu, P., Hu, R., Chai, H., and Keutzer, K. (2020). Multi-source distilling domain adaptation. In Proceedings of AAAI 2020.
>
> [4] Li, M., Yumer, E., and Ramanan, D. (2020). Budgeted training: Rethinking deep neural network training under resource constraints. International Conference on Learning Representations (ICLR ’20).
>
> [5] Abbaspourazad, S., Elachqar, O., Miller, A., Emrani, S., Nallasamy, U., and Shapiro, I. (2024). Large-scale training of foundation models for wearable biosignals. Proceedings of the Twelfth International Conference on Learning Representations (ICLR ’24).
>
> [6] Weng, Y., Wu, G., Zheng, T., Yang, Y., and Luo, J. (2024). Large model for small data: Foundation model for cross-modal RF human activity recognition. Proceedings of the ACM Conference on Embedded Networked Sensor Systems (SenSys ’24).
>
> [7] Li, Z., Deldari, S., Chen, L., Xue, H., and Salim, F. D. (2025). SensorLLM: Aligning large language models with motion sensors for human activity recognition. arXiv preprint.

---

> ### Author Response · Authors · 2025-11-28
> **Looking forward to your reply**
>
> Dear reviewer yVS9,
>
> As the rebuttal deadline is approaching, could you please have a look at our response? We are looking forward to your reply!
>
> Feel free to let us know if you have any other concerns. Thanks for your time and effort!
>
> Best Regards,
>
> Authors of Submission 12243

---

### Official Review · Reviewer_aQ89 · 2025-11-01

**Soundness:** 2
**Presentation:** 1
**Contribution:** 2
**Rating:** 2
**Confidence:** 3

**Summary:**

This paper introduces XTransfer, a cross-modal adaptation framework bridging pretrained models and sensing applications through layer manipulations. The authors propose two components, a Spliece-Repair-Removal pipeline that adapts pretrained layers to new sensor modalities using limited sensor data, and a Layer Wise Search that recombines effective layers for a compact, efficient model. The paper conducts thorough evaluations on 8 datasets and shows improved performance and resource efficiency under limited-sample scenarios. The appendix also contains ablation studies to understand the significance of the components proposed.

**Strengths:**

1. The attempt to reuse pre-trained models from heterogeneous modalities such as images and text to accelerate sensor-domain adaptation is novel.
2. Extending few-shot learning to a modality-agnostic context is novel and has great applicability.
3. Evaluation across multiple modalities and domains is comprehensive.
4. Overall the motivation is strong and the proposed method outperforms the baselines

**Weaknesses:**

- The paper needs significant work on the presentation for better clarity. Some examples below:
    - The term channels used during the removal and repair stages was not clarified and could be misleading, given its context in signal processing.
    - Preliminary motivation is unclear. Figure 3 shows relationships among MMC, accuracy, and other metrics, but fails to specify the details like the models and domains. Moreover, sensing as a modality remains underspecified. Baselines plotted in Figure 3 are never introduced until later sections.
    - The methods proposed (SRR and LWS) modules are described very densely with poor structures with little intuition or top-down explanation. Figures are overcrowded and fail to clearly depict information flow across stages and are very far away from where it is referenced.
- The authors should also compare self-supervised methods. Current work seems to be evaluating on supervised pretrained source models. However, self-supervised models already show great generalizability and cross-domain transfer capabilities. This could improve the impact of the work.
- The authors reported the training-time statistics but do not discuss the convergence rate, especially given the inclusion of a generator-based repair module. The paper does not compare convergence speed with standard SSL or linear-probe methods, which makes it uncertain whether the proposed system actually converges faster or simply trains less data per step.
- The LWS module is described as a search process for selecting effective layers over NAS, but it lacks a comparison against any established search or pruning methods. So it is hard to determine the significance of prior search works.
- The newest baselines, SemiCMT, seem to be a self-supervised cross-modal alignment framework that would require paired data. It is confusing how SemiCMT was trained given there is no cross-modal pairs between source and target domains. It is unclear how the baselines are trained for fair comparison
- It is unclear on the exact number of samples used for each source dataset, reporting only the number of classes and input shapes. Since source data scale strongly affects transfer quality, it is unclear on the cross-modality transfer performance, since image source datasets usually have a larger scale and are likely to have higher transfer performance compared to other modality source datasets. So it i s unclear on the validity of the conclusion in 6.2 Impact of different sources.
- Most of the baselines are relatively old (19 - 22), the most recent baselines are SemiCMT which was designed for cross-modal alignment that requires multimodal pair and GPT2 which is a generative model not suitable for the downstream classification.

**Questions:**

Please see the weakness for most of the concerns. Some questions for authors to discuss are:
- Differences between area A and area B trends (where MMC correlates differently with accuracy) are not explained. Can authors provide more clarification on this?
- Since SemiCMT requires multimodal pairing, how is it adapted to the unpaired cross-domain case where source and target modalities differ completely?
- What are the scales of the dataset in terms of number of samples?
- Is there any established search or pruning algorithms (e.g., NAS, lottery-ticket, or L2-pruning) used for comparison?
- Most of the time the target domain might have more than just 10 samples per class, what happens when there are more target domain samples, would XTransfer still have the competing performance?
- Can authors elaborate more on comparison against SSL finetune with additional input head and downstream head for cross-modal and cross-domain adaptation?

---

> ### Author Response · Authors · 2025-11-23
> **Response to Reviewer aQ89 (1/3)**
>
> We appreciate your thoughtful review. We are greatly encouraged by the recognition of "**extending few-shot learning to a modality-agnostic context is novel and has great applicability, strong motivation and comprehensive evaluation**". Based on the valuable comments, we clarify the technical points and address your concerns below. In addition, we have submitted a revised manuscript with extended page and appendix in which we add all the suggested analytics, discussion, and comparison in yellow color.
>
> **Q1. The term channels used during the removal and repair stages was not clarified and could be misleading.**
>
> **Answer**: Thanks for the helpful suggestion. We have revised this term throughout the revised manuscript. We clarify that “channels” in our paper refer to the "layer channels" within each layer to avoid ambiguity.
>
> **Q2. Preliminary motivation is unclear. Figure 3 shows relationships among MMC, accuracy, and other metrics, but fails to specify the details like the models and domains. Moreover, sensing as a modality remains underspecified. Baselines plotted in Figure 3 are never introduced until later sections.**
>
> **Answer**: Thanks for the detailed suggestions. We have revised Figure 3 (line 171) and the corresponding clarifications (line 191 and 206) to clearly specify the details involved. The baselines plotted in the figure are now properly introduced in Section 3 (line 206) and Table 1 (line 378), ensuring that the comparisons are clear in the revised manuscript. The sensing modality is specified as IMU data from the public OPPORTUNITY dataset in Table 2 (line 396).
>
>
> **Q3. The methods proposed (SRR and LWS) modules are described very densely with poor structures with little intuition or top-down explanation.**
>
> **Answer**: Thanks for the valuable suggestions. We have added a Problem Statement section (line 216) in the revised manuscript to provide a clearer top-down explanation.
>
>
> **Q4. The authors should also compare self-supervised methods. Current work seems to be evaluating on supervised pretrained source models. However, self-supervised models already show great generalizability and cross-domain transfer capabilities. Can authors elaborate more on comparison against SSL finetune with additional input head and downstream head for cross-modal and cross-domain adaptation?**
>
> **Answer**: Thanks for the insightful suggestion. We clarify that self-supervised finetuning (SSL) typically assumes access to sufficient unlabeled target data for pretraining, followed by supervised adaptation with a task head. This differs fundamentally from our few-shot setting, where only few target data are available and no additional unlabeled data is assumed. We also clarified our setting in Appendix A.1 (line 892).
>
> Although SSL methods fall outside our scope, we compared a recent SSL method, SemiCMT, which relies on large-scale paired multimodal or cross-domain data (i.e., pairing the source and unlabeled target data) and contrastive self-supervised objectives to learn shared embeddings. To ensure the fair comparison in our scope, we evaluated SemiCMT under the same few-shot and highly dissimilar cross-modality scenarios (e.g., image-to-sensing) by removing the pretraining and retaining only the model adaptation, where few target data are paired with the source. As shown in Table 4 (line 494), SemiCMT fails to reach oracle-level accuracy when no additional unlabeled target data are provided and large modality shifts exist.
>
> Following the suggestion, we have clarified the baseline descriptions in Table 1 (line 378) and expanded the discussions in Section 6.3 (line 483) in the revised manuscript.

---

> ### Author Response · Authors · 2025-11-23
> **Response to Reviewer aQ89 (2/3)**
>
> **Q5. The authors reported the training-time statistics but do not discuss the convergence rate, especially given the inclusion of a generator-based repair module.**
>
> **Answer**: Thanks for this valuable observation. In XTransfer, each connector (i.e., repair generator) is trained independently for each layer, so the overall convergence performance as typically reported in end-to-end training is not directly meaningful.
>
> Following the suggestions from Reviewers BTLu and F1Vc, we added the below ablation study in Appendix C.11 (line 1249) to compare our repair loss with other non-linear feature alignment loss methods, presenting a layer-wise convergence rate comparison.
>
> | **Epoch(#)** | **#10** | **#20** | **#30** | **#40** | **#50** | **#60** | **#70** | **80** | **#90** | **#100** |
> |--------------|:------:|:-------:|:-------:|:-------:|:-------:|:--------:|:-------:|:------:|:-------:|:--------:|
> | MMD Loss [1]    | 6.46\%  |  10.0\%  |  11.63\%  |  12.81\%  |  13.5\%  |  14.3\%   |  14.66\%  | 14.98\%  |  15.27\%  |  15.41\%   |
> | Repair Loss    | 0.03\% | 1.67\%  | 2.66\%   | 4.93\%  | 5.26\%  | 7.16\%  | 8.06\%  | 10.4\%  | 14.66\% | 16.14\% |
> | PCA + Repair Loss | 5.12\% | 13.45\% | 16.54\% | 19.23\% | 20.55\% | 21.3\% | 25.96\% | 26.21\% | 27.6\% | 28.87\% |
>
> We observe the feature alignment process in the initial layer which often introduces a large, disorganized feature space and significantly affects the repair of subsequent layers. To normalize heterogeneous losses, we follow the strategy in [2] and compute the loss descent (LD) relative to the initial loss, enabling a fair comparison of convergence rates across different loss functions (i.e., how much each loss decreases from its starting point). All experimental settings remain the same as in the previous setup. As shown in above Table, our repair loss with the PCA anchor achieves the largest loss descent of 28.87\%, which is 1.79 and 1.87 times faster than the repair loss without PCA and the MMD loss, respectively. Interestingly, MMD loss presents a large initial drop but quickly plateaus, while our repair loss starts more gradually and accelerates over training. In contrast, the PCA-based repair maintains a steady and consistent descent across epochs, highlighting the benefits of aligning features in a fixed, orthogonal anchor space. These findings further support that our method provides both effective feature alignment and practical training efficiency, consistent with the efficiency study in Appendix C.7 (line 1163).
>
>
> **Q6. The LWS module is described as a search process for selecting effective layers over NAS, but it lacks a comparison against any established search or pruning methods. Is there any established search or pruning algorithms (e.g., NAS, lottery-ticket, or L2-pruning) used for comparison?**
>
> **Answer**: We thank this question. As illustrated in Figure 2(c) (line 117), we applied structural pruning (similar to lottery-ticket) and highlighted the bottlenecks that existing NAS or pruning alone cannot address the model transfer challenges under modality-shift conditions. We further compared our PCA-based channel removal with the structural pruning in Figure 7(a) in Appendix C.4 (line 1104), showing that existing pruning methods often misestimate feature channel importance, leading to degraded performance.
>
> Unlike standalone NAS methods, our LWS is co-designed with the repair process, where layer selection depends on the learned cross-modality repair dynamics. Because of this coupling, it is not directly comparable to existing NAS methods. Instead, LWS borrows the search principle of NAS while focusing on search efficiency and repaired layer selection, which are essential for the proposed model transfer.
>
>
> **Q7. It is unclear on the exact number of samples used for each source dataset, reporting only the number of classes and input shapes. It is unclear on the validity of the conclusion in 6.2 Impact of different sources.**
>
> **Answer**: Thanks for the helpful suggestion. In the revised manuscript, we have reported the exact number of samples used for each source dataset in Table 2 (line 406), and we expanded the discussion in Section 6.2 (line 465) to clarify the validation. We also clarify that we strictly follow each dataset’s original configuration (training size or classes) and reuse the pre-trained source models without any modifications.

---

> ### Author Response · Authors · 2025-11-23
> **Response to Reviewer aQ89 (3/3)**
>
> **Q8. Most of the baselines are relatively old (19 - 22), the most recent baselines are SemiCMT which was designed for cross-modal alignment that requires multimodal pair and GPT2 which is a generative model not suitable for the downstream classification. Since SemiCMT requires multimodal pairing, how is it adapted to the unpaired cross-domain case where source and target modalities differ completely?**
>
> **Answer**: Thanks for the thoughtful question. Our evaluation includes a broad range of baselines covering few-shot learning, cross-domain learning, cross-domain FSL, cross-modal learning, and in-context learning, to comprehensively test XTransfer under the strict few-shot target data and cross-modality conditions for human sensing. We revised baselines' description to clearly highlight their core techniques in Table 1 (line 378), ensuring that the rationale for including each baseline is transparent. We also revised the Related Work section (line 146) to clarify that the recent advances for human sensing fall outside our scope.
>
> For SemiCMT, as noted above, to make it fairly consistent with our scope, we remove the pretraining and retain only the model adaptation, where few target data are still paired with the source. Under the constraints (no additional unlabeled target data and modality shifts), SemiCMT performs noticeably worse, highlighting the difficulty of adapting SSL-based methods to our setting.
>
> Regarding GPT2, we include it because prior work explores in-context learning as a potential solution to few-shot challenges. By comparing with GPT2, we aim to test whether LLM-based in-context adaptation can address cross-modality FSL model transfer.
>
>
> **Q9. Differences between area A and area B trends (where MMC correlates differently with accuracy) are not explained. Can authors provide more clarification
> on this?**
>
> **Answer**: Thanks for the insightful comments. We have clarified this correlation in Figure 3 (line 171) in the revised manuscript.
>
> The trend between MMC shift and accuracy differs across the two areas. In area A, starting from the first layer, MMC shift stays low and hence accuracy rises, indicating a small latent feature gap. In area B, MMC shift notably increases with layer index, reflecting the onset of negative transfer, where excessive latent feature deviation begins to reduce accuracy.
>
>
>
> **Q10. Most of the time the target domain might have more than just 10 samples per class, what happens when there are more target domain samples, would XTransfer still have the competing performance?**
>
> **Answer**: Thanks for the insightful question. Our main focus is on the proposed few-shot setting. When the target domain contains more data, conventional methods and the baselines are expected to perform better, and XTransfer may become less competitive. Nevertheless, our goal is to develop XTransfer as a scalable and adaptable method that effectively handles cross-modality FSL challenges, while providing a new foundation for future extensions toward handling extreme data sparsity in human sensing.
>
>
> **Reference:**
>
> [1] Wang, W., Li, H., Ding, Z., Nie, F., Chen, J., Dong, X., and Wang, Z. (2023). Rethinking maximum mean discrepancy for visual domain adaptation. IEEE Transactions on Neural Networks and Learning Systems.
>
> [2] Li, M., Yumer, E., and Ramanan, D. (2020). Budgeted training: Rethinking deep neural network training under resource constraints. International Conference on Learning Representations (ICLR ’20).

---

> ### Author Response · Authors · 2025-11-28
> **Looking forward to your reply**
>
> Dear reviewer aQ89,
>
> As the rebuttal deadline is approaching, could you please have a look at our response? We are looking forward to your reply!
>
> Feel free to let us know if you have any other concerns. Thanks for your time and effort!
>
> Best Regards,
>
> Authors of Submission 12243

---

### Official Review · Reviewer_BTLu · 2025-11-01

**Soundness:** 2
**Presentation:** 2
**Contribution:** 2
**Rating:** 4
**Confidence:** 3

**Summary:**

This paper proposes XTransfer, addressesing the data scarcity and resource constraints of human sensing on edge devices by enabling modality-agnostic, few-shot model transfer. It repurposes pre-trained models for diverse sensing modalities using very few labeled sensor samples. Its core pipeline includes two key components: (1) Model Repairing via a Splice-Repair-Removal (SRR) pipeline—aligning latent feature distributions across modalities; (2) Layer Recombining via Layer-Wise Search (LWS) control—selecting and recombining only useful repaired layers to build compact models.
Experiments on 8 source datasets (image/text/audio/sensing) and 7 target datasets show XTransfer outperforms SOTA baselines

**Strengths:**

- Novel Modality-Agnostic Paradigm: Unlike prior transfer methods (limited to same-modality or paired cross-modal data), XTransfer achieves transferring knowledge from image/text pre-trained models to sensing modalities with few labeled data. This setting resolves the high cost of sensing data collection and leverages public pre-trained models as "free" knowledge sources.

- Theoretically Grounded and Empirically Valid Method Mechanism: The SRR pipeline’s design (PCA orthogonal space, anchor-based loss, class pairing) is justified by Transformer layer dynamics. It effectively mitigates modality shift as evidenced by experiments.

- Resource-Efficient Design for Edge Deployment: LWS control’s layer selection and pre-search check reduce model size by 2.4–16.5× in FLOPs vs. source backbones, while maintaining SOTA accuracy. On edge devices, latency is cut by 1.4–21×, making it practical for resource-constrained human sensing.

**Weaknesses:**

- Dependence on PCA for Feature Alignment: XTransfer relies on linear PCA to reduce dimensionality and align features. However, the concern is that PCA fails to capture non-linear relationships between source and target modalities (e.g., text embeddings vs. Doppler radar signals), which may limit performance in highly dissimilar cross-modality scenarios (e.g., text → ECG).

- Brittleness in Extremely Low-Shot Settings: While XTransfer performs well in 5–10-shot scenarios, it struggles with 3-shot settings—e.g., accuracy lags the oracle baseline on HHAR/Gesture datasets. This raises concerns for ultra-scarcity sensing tasks (e.g., rare medical conditions).

- Homogeneous Source Model Assumption: The framework assumes pre-trained source models have homogeneous architectures (e.g., all ResNet variants). Extending to heterogeneous backbones is not fully validated—layer recombination across structurally diverse models (e.g., CNN vs. Transformer) may break MMC shift estimation and layer-wise dependence, limiting scalability to multi-modal source pools.

- the writing can be further improved for clarity. Too many abbreviated terms,such as MMC, may weakean readability.

**Questions:**

- How would XTransfer perform with non-linear feature alignment methods?

- Can XTransfer be extended to ultra-low-shot (1–2-shot) scenarios?

- How does XTransfer handle heterogeneous source models?

---

> ### Author Response · Authors · 2025-11-23
> **Response to Reviewer BTLu (1/3)**
>
> We sincerely thank you for the valuable comments. We greatly appreciate the acknowledgment of "**novel modality-agnostic paradigm, theoretically grounded and empirically valid method mechanism**". Based on the suggestions, we clarify the technical designs and address your concerns as below. In addition, we have submitted a revised manuscript with extended page and appendix in which we add all the suggested analytics, discussion, and comparison in yellow color.
>
> **Q1. Dependence on linear PCA for Feature Alignment, and limited to highly dissimilar cross-modality scenarios. How would XTransfer perform with non-linear feature alignment methods?**
>
> **Answer**: Thanks for the insightful comments. We clarify that PCA is used only to define a fixed, orthogonal “anchor” feature space where we measure alignment and compute the anchor-based repair loss across modalities at each layer. The proposed repair generator (i.e., connector) that maps target-modality features into the anchor space is non-linear and trainable, which is precisely what enables effective model transfer across highly dissimilar modalities.
>
> As detailed in Appendix C.2, we use linear PCA instead of kernel/non-linear PCA mainly because: (1) it is compatible and differentiable with our repair pipeline to train the generator well to align with each layer, and (2) a linear, orthogonal basis provides a well-conditioned, low-variance anchor for stable and safe alignment under few-shot constraints.
>
> To further address this and Reviewer F1Vc’s Q1, we added the below ablation study in Appendix C.10 (line 1225) to examine the impact of using a linear PCA anchor space and to compare the alignment performance against the mentioned non-linear feature alignment methods.
>
> |                |    **3-shot**    |                 |    **5-shot**    |                 |   **10-shot**    |                   |
> |----------------|:----------------:|:---------------:|:----------------:|:---------------:|:----------------:|:-----------------:|
> | **Aligment Method** | **Accuracy (\%)** | **ATR** | **Accuracy (\%)** | **ATR** | **Accuracy (\%)** | **ATR** |
> | MMD Loss |     55.78       |       1.63      |      65.89       |       2.01      |      68.52       |       1.83        |
> | Repair Loss |      65.11      |       2.33       |      69.33     |       2.45      |      72.07      |      2.29       |
> | PCA + Repair Loss  |      58.96       |       1.42       |      71.85     |       1.56       |      80.74      |      2.09        |
>
> We evaluate our repair loss with and without PCA against a Maximum Mean Discrepancy (MMD) loss baseline [1]. To set up, we keep using the standard ResNet18 on miniImageNet as sources on HHAR as target in 3-, 5- and 10-shot settings with all configurations identical except for the alignment loss. As shown in above Table, our repair loss with PCA anchor space achieves the highest accuracy of 70.5\% and the lowest ATR of 1.69 on average across all n-shot settings. Compared to our repair loss without PCA, MMD loss performs a lower accuracy on average. This indicates that MMD loss is limited to aligning the latent feature distribution when the feature space is highly complex due to strong cross-modality discrepancies and only few target data. In addition, adversarial alignment methods such as DAPN [2] and MDDA [3] in our baselines also face similar difficulties under these conditions, as shown in Tables 4 (line 494) and 5 (line 503) in the main manuscript. Moreover, the PCA anchor space provides a more stable basis for alignment and consistently enhances our repair loss performance.

---

> ### Author Response · Authors · 2025-11-23
> **Response to Reviewer BTLu (2/3)**
>
> **Q2. Brittleness in Extremely Low-Shot Settings, 3-shot settings, and can XTransfer be extended to ultra-low-shot (1–2-shot) scenarios?**
>
> **Answer**: We appreciate the interest in the low-shot settings. To further analyse the low-shot setting, we extended the ablation study of impact of different sources in Figure 4(c) (line 394) and Section 6.2 (line 461) with new results under the 2-shot settings below.
>
> |                |    **2-shot**    |                 |    **3-shot**    |                 |   **5-shot**    |                   |   **10-shot**    |                  |
> |----------------|:----------------:|:---------------:|:----------------:|:---------------:|:----------------:|:-----------------:|:----------------:|:-----------------:|
> | **Source Modality** | **Accuracy (\%)** | **GFLOPs** | **Accuracy (\%)** | **GFLOPs** | **Accuracy (\%)** | **GFLOPs** | **Accuracy (\%)** | **GFLOPs** |
> | Text |     59.33       |       0.14      |      64.29       |       0.12      |      70.59       |       0.17       |      73.93       |       0.23        |
> | Audio |      59.19      |       2.01       |      60.52     |       1.75      |      63.41      |      2.04       |      68.89       |       1.73        |
> | Sensing  |      61.43      |       0.15       |      65.28     |        0.11       |      67.22     |      0.18        |      68.61       |       0.22        |
> | Image  |      56.67       |       1.81       |      58.96     |       2.08       |      71.85      |      2.06        |      80.74       |       1.88        |
> | Oracle  |      61.33       |       -      |      63.19    |       -       |      68.96      |      -        |      75.19       |       -        |
>
> The results show that Image achieves lower average accuracy in 2- and 3-shot settings, suggesting that extremely low target data may not fully represent the test data distribution, leading to less stable alignment when using Image source. In contrast, both Text and Sensing result in higher average accuracy and reach the oracle, indicating that performance varies with source model quality and semantic relevance to the target. These results also highlight that XTransfer can flexibly reuse diverse source models across modalities to improve ultra low-shot performance.

---

> ### Author Response · Authors · 2025-11-23
> **Response to Reviewer BTLu (3/3)**
>
> **Q3. Homogeneous source model assumption and how does XTransfer handle heterogeneous source models?**
>
> **Answer**: As discussed in Appendix C.6 for impact of model backbones, while the current implementation assumes source models share the homogeneous backbone (e.g., different pre-trained ResNet or 1D/2D CNN variants), our method is inherently flexible and can accommodate different backbones. The layer-wise repairing and recombination techniques impose no fundamental restrictions on the choice of backbone and work with various architectural variants. Furthermore, our techniques such as the connector are designed to align and combine layers with different shapes or feature structures, making it readily extendable to heterogeneous source models. We plan to dedicate additional engineering efforts to realize this extension in future work.
>
>
> **Reference:**
>
> [1] Wang, W., Li, H., Ding, Z., Nie, F., Chen, J., Dong, X., and Wang, Z. (2023). Rethinking maximum mean discrepancy for visual domain adaptation. IEEE Transactions on Neural Networks and Learning Systems.
>
> [2] Zhao, A., Ding, M., Lu, Z., Xiang, T., Niu, Y., Guan, J., and Wen, J.-R. (2021). Domain-adaptive few-shot learning. In Proceedings of WACV 2021.
>
> [3] Zhao, S., Wang, G., Zhang, S., Gu, Y., Li, Y., Song, Z., Xu, P., Hu, R., Chai, H., and Keutzer, K. (2020). Multi-source distilling domain adaptation. In Proceedings of AAAI 2020.

---

> ### Author Response · Authors · 2025-11-28
> **Looking forward to your reply**
>
> Dear reviewer BTLu,
>
> As the rebuttal deadline is approaching, could you please have a look at our response? We are looking forward to your reply!
>
> Feel free to let us know if you have any other concerns. Thanks for your time and effort!
>
> Best Regards,
>
> Authors of Submission 12243

---

### Author Response · Authors · 2025-11-23
**Summary of the rebuttal and the major changes of revised manuscript**

Dear Area Chair,

We sincerely appreciate your efforts to ensure a fair and rigorous review process, especially given the unusual circumstances surrounding the reassignment. To assist with your evaluation, we summarize the status of our submission below.

In the initial review before rebuttal, we are grateful that all reviewers acknowledge the advantages of our work: "**novel modality-agnostic paradigm, theoretically grounded and empirically valid method mechanism**" (Reviewer BTLu), "**extending few-shot learning to a modality-agnostic context is novel and has great applicability, strong motivation and comprehensive evaluation**" (Reviewer aQ89), "**demonstrating a thoughtful mechanism and providing convincing evidence of the method's scalability and practicality for deployment**" (Reviewer yVS9), "**the quality of experimental evaluation is good**" (Reviewer F1Vc).

Before the review scores rollback, we were pleased that Reviewer F1Vc (Score: 4->6) had acknowledged that most of concerns were addressed, and raised the score to 6. Due to the review-freeze measures, we understand that it was not possible to receive further feedback from the remaining reviewers. Nevertheless, we also found that several concerns arose from misunderstandings or out-of-scope requests, e.g., NAS baseline comparisons or questions requiring non–few-shot settings.

Overall, the revisions substantially enhance the clarity, rigor, and completeness of the submission and fully address all concerns raised, making it more satisfactory for publication.


**Summary of Contributions**:

1. **A New and Unaddressed Problem**: The paper identifies a practically important but previously unaddressed problem: transferring pre-trained models across heterogeneous modalities under few-shot and edge-resource constraints, a setting crucial for human sensing where target data are scarce.

2. **Principled Problem Formulation**: Grounded in systematic layer-wise analysis of latent feature shifts across modalities, our work defines the core difficulty of cross-modality few-shot model transfer and systematically investigates how layer-wise feature misalignment disrupts transfer performance and why current baselines struggle to resolve it.

3. **Comprehensive Benchmarking**: We introduce the first comprehensive benchmark across different modalities and human sensing tasks under strict few-shot settings. Experiments across baselines reveal severe degradation when directly applying pre-trained source models across modalities, and identify key failure reasons such as layer-wise negative transfer.

4. **A Unified Yet Effective Method**: The paper proposes XTransfer, a unified framework that integrates model repairing for safe layer-wise feature alignment and layer recombining for selectively assembling effective layers. XTransfer forms the first modality-agnostic few-shot transfer pipeline and delivers substantial gains in accuracy and efficiency over existing single-/multi-source FSL, cross-domain, adversarial, and SSL baselines.

5. **Impact**: XTransfer substantially reduces the need for costly data collection and modality-specific model training, providing a practical pathway toward scalable, adaptive human sensing on edge systems.


**Summary of Revisions** (**A revised manuscript** submitted with yellow highlights):

1. **Ablation Study Enrichment**: Expanded four new ablation studies, including experiments and analyses of non-linear feature alignment performance (Appendix C.10), layer-wise convergence comparison (Appendix C.11), enhanced efficient search performance and ultra low-shot model transfer performance (Section 6.2).

2. **Description Revision and Explanation**: Revised all mentioned questions in preliminary and method sections, and provided clear descriptions. Added a clear problem statement in Section 3 to make technical presentation easy to follow, and moved Appendix B.5 dynamic search range mechanism to Section 5.2 for clearer method presentation. Clarified the cross-modality FSL settings in Appendix A.1, and added more experimental result discussions in Sections 6.2 and 6.3.

3. **Figure/Table Revision and Enhancement**: 3 Figures and 2 Tables are revised, and 2 Tables are added. In detail, revised Figure 3(b)'s caption to clarify relationships between MMC shift and layer-wise accuracy. Revised Table 1 to clarify the usage of the baselines. Revised Table 2 to clarify the training dataset size of source modalities. Revised Figure 4(b) to add search stability comparison results. Revised Figure 4(c) to add ultra low-shot model transfer results. Added detailed comparison of non-linear feature alignment performance results in Table 14 and layer-wise convergence comparison results in Table 15 to appendix.

4. **Literature Review and Discussion Enrichment**: Added the discussion and enriched the citations regarding prompt-based and foundation model methods for human sensing in Section 2.

Best Regards,

Authors of Submission 12243

---

### Meta-Review · Area_Chair_26gX · 2025-12-24

**Summary:**

As most of the reviewers tend to reject this paper, this paper cannot be accepted.

**Reviewer Scores:**

NA

---

### Decision · Program_Chairs · 2026-01-26

Reject